# Open Problems in Mechanistic Interpretability

**Lee Sharkey**[*]                                                       *Apollo Research*

**Bilal Chughtai**[*]                                                     *Apollo Research*

**Joshua Batson**                                                         *Anthropic*

**Jack Lindsey**                                                          *Anthropic*

**Jeff Wu**                                                               *Anthropic*[†]

**Lucius Bushnaq**                                                        *Apollo Research*

**Nicholas Goldowsky-Dill**                                              *Apollo Research*

**Stefan Heimersheim**                                                    *Apollo Research*

**Alejandro Ortega**                                                      *Apollo Research*

**Joseph Bloom**                                                          *Decode Research*

**Stella Biderman**                                                       *Eleuther AI*

**Adria Garriga-Alonso**                                                  *FAR AI*

**Arthur Conmy**                                                          *Google DeepMind*

**Neel Nanda**                                                            *Google DeepMind*

**Jessica Rumbelow**                                                      *Leap Laboratories*

**Martin Wattenberg**                                                     *Harvard University*

**Nandi Schoots**                              *King's College London and Imperial College London*

**Joseph Miller**                                                         *MATS*

**William Saunders**                                                      *METR*

**Eric J. Michaud**                                                       *MIT*

**Stephen Casper**                                                        *MIT*

**Max Tegmark**                                                           *MIT*

**David Bau**                                                             *Northeastern University*

**Eric Todd**                                                             *Northeastern University*

**Atticus Geiger**                                                        *Pr(AI)$^2$r group*

**Mor Geva**                                                              *Tel Aviv University*

**Jesse Hoogland**                                                        *Timaeus*

**Daniel Murfet**                                                         *University of Melbourne*

**Tom McGrath**                                                           *Goodfire*

**Reviewed on OpenReview:** *https://openreview.net/forum?id=91H76m9Z94*

---

[*]Correspondence to: leedsharkey@gmail.com and brchughtaii@gmail.com

[†]Work done prior to joining Anthropic.

## Abstract

Mechanistic interpretability aims to understand the computational mechanisms underlying neural networks' capabilities in order to accomplish concrete scientific and engineering goals. Progress in this field thus promises to provide greater assurance over AI system behavior and shed light on exciting scientific questions about the nature of intelligence. Despite recent progress toward these goals, there are many open problems in the field that require solutions before many scientific and practical benefits can be realized: Our methods require both conceptual and practical improvements to reveal deeper insights; we must figure out how best to apply our methods in pursuit of specific goals; and the field must grapple with socio-technical challenges that influence and are influenced by our work. This forward-facing review discusses the current frontier of mechanistic interpretability and the open problems that the field may benefit from prioritizing.

# Contents

# 1 Introduction

Recent progress in artificial intelligence (AI) has resulted in rapidly improved AI capabilities. These capabilities are not designed by humans. Instead, they are learned by deep neural networks (Hinton et al., 2006; LeCun et al., 2015). Developers only need to design the training process; they do not need to – and in almost all cases, do not – understand the neural mechanisms underlying the capabilities learned by an AI system.

Although human understanding of these mechanisms is not necessary for AI capabilities, understanding them would enhance several *human* abilities. For example, it would permit better human control over AI behavior and better monitoring during deployment. It would also facilitate trust in AI systems, allowing us to fully realize their potential benefits by enabling their deployment in safety-critical and ethically-sensitive settings.

Beyond the engineering benefits, understanding AI systems offers immense scientific opportunities. For the first time in history, we can create and study artificial minds with a level of access and control that is simply not possible in biological systems. What new laws of nature governing the mechanisms of minds might we discover from studying the internal workings of AI systems? What new methods for analyzing biological neural systems might neuroscientists be able to glean from understanding artificial ones?

The scientific opportunities are not limited to the field of AI. If an AI can outperform tools designed by humans in a given scientific field, it suggests that the AI system is representing something about the world currently unknown to us. We can develop a deeper comprehension of the world by understanding those representations. What can we learn about protein folding from AIs that can successfully predict protein structure? What insights can we glean about disease from a radiographer that performs beyond human ability?

*Mechanistic interpretability* might unlock these benefits. This field of study aims to understand neural networks' decision-making processes. Here, we define "Understanding a neural network's decision-making process" as the ability to use knowledge about the mechanisms underlying a network's decision-making process in order to successfully predict its behavior (even on arbitrary inputs) or to accomplish other practical goals with respect to the network. Such goals might include more precise control of the network's behavior, or improved network design. Interpretability promises greater assurances for AI systems through a better understanding of what neural networks have learned, thus enabling us to realize their potential benefits.

## 1.1 The focus of this review: Open problems and the future of mechanistic interpretability

Several recent reviews of mechanistic interpretability research and related topics exist (Rauker et al., 2023; Geiger et al., 2022; Bereska & Gavves, 2024; Ferrando et al., 2024; Rai et al., 2024; Anwar et al., 2024; Davies & Khakzar, 2024; Mosbach et al., 2024; Mueller et al., 2024). Our review takes a more forward-looking stance. We discuss not only where the frontier is today, but also which directions we might benefit most from prioritizing in the future.

### 1.1.1 Why 'mechanistic' interpretability?

The distinction between interpretability and mechanistic interpretability is not always clear and is therefore worth clarifying. The motivations and methods used in interpretability work are often diverse (Lipton, 2018; Doshi-Velez & Kim, 2017; Jacovi, 2023). As a result, there are many ways in which interpretability research might be categorized. Prior categorizations of interpretability include causal vs. correlational methods, supervised vs. unsupervised methods, bottom-up vs. top-down methods, among others (Geiger et al., 2024b; Mueller et al., 2024; Bereska & Gavves, 2024; Belinkov, 2022a; Zou et al., 2023a; Davies & Khakzar, 2024). This review focuses specifically on *mechanistic* interpretability. But what distinguishes 'mechanistic interpretability' from interpretability in general? It has been noted that the term is used in a number of (sometimes inconsistent) ways (Saphra & Wiegreffe, 2024). In this review, we use the term 'mechanistic interpretability' in a technical sense, referring specifically to work that investigates the mechanisms underlying neural network generalization. Mechanistic interpretability represents one of three threads of interpretability research, each with distinct but sometimes overlapping motivations, which roughly reflects the changing aims of interpretability work over time.

The first thread aims to build AI systems that are interpretable by design. Much early interpretability work focused on explaining the sensitivity of machine learning models to inputs and training data. This work typically used small-to-medium sized models designed to be easily interpretable, such as decision trees (Breiman, 1984; Hu et al., 2019), linear models (Roweis & Ghahramani, 1999), and generalized additive models (Hastie & Tibshirani, 1986; Agarwal et al., 2021). These models could be used alongside attribution methods such as influence functions (Hampel, 1974; Koh & Liang, 2017) and Shapley values (Shapley, 1997; Lundberg & Lee, 2017), which were common techniques used to characterize model decision boundaries with respect to inputs. "Interpretability by design" continues to be an active research area, including architectures such as Concept-Bottleneck Models (Koh et al., 2020), Backpack Language Models (Hewitt et al., 2023), Kolmogorov-Arnold Networks (Liu et al., 2024c), and sparse decision trees (Xin et al., 2022).

With the rise of larger-scale nonlinear neural networks (Krizhevsky et al., 2012; He et al., 2016), another thread grew in importance, driven primarily by the question: Why did my model make this particular decision? However, one challenge in interpreting larger networks was finding attribution methods that could scale to large networks (Zeiler & Fergus, 2014). In response, a number of local attribution methods were developed, including grad-CAM (Selvaraju et al., 2019), integrated gradients (Sundararajan et al., 2017), and masking-based causal attribution (Fong & Vedaldi, 2017), SHAP (Lundberg & Lee, 2017), LIME (Ribeiro et al., 2016), and many other methods, including backprop-based visualization methods (Simonyan et al., 2014a; Nguyen et al., 2016b).

Inspired by, for example, Inception (Szegedy et al., 2015) and GPT-3 (Brown et al., 2020), another thread emerged as models became capable of more profound generalization. Focused on the broader subject of generalization, it was driven by the question: How did my model solve this general class of problems? Due to its emphasis on the mechanisms underlying neural network generalization, the work in this category is commonly referred to as 'mechanistic interpretability' (in addition to other, more cultural reasons, according to (Saphra & Wiegreffe, 2024)). This kind of interpretability work is driven by a fundamental hypothesis in deep learning that generalization arises from shared computation (LeCun et al., 2015). Early work in this area, such as feature visualization (Olah et al., 2017b), or network dissection (Bau et al., 2020), sought "global" explanations for model generalization by investigating the roles of model components across a class of decisions. More recent work in this area looks at "circuits" of components (Wang et al., 2023), generalizable patterns of information flow (Geva et al., 2023), representation subspaces (Geiger et al., 2024c; Zou et al., 2023a) and probes (or self-supervised searches via sparse dictionary learning) for representation vectors that carry information that generalizes across many instances for a particular task or set of tasks (Huben et al., 2024; Bricken, 2023; Todd et al., 2024; Hollinsworth et al., 2024).

## 1.2 Types of open problems

The field of mechanistic interpretability ultimately aims to achieve concrete scientific and engineering goals. For instance, we would like to be able to:

- Monitor AI systems for signs of cognition related to dangerous behavior (Section 3.2);

- Modify internal mechanisms and edit model parameters to adapt their behavior to better suit our needs (Section 3.2.2);

- Predict how models will act in unseen situations or predict when a model might learn specific abilities (Section 3.3);

- Improve model inference, training, and mechanisms to better suit our preferences (Section 3.4);

- Extract latent knowledge from AI systems so we can better model the world (Section 3.5).

Despite recent hopeful signs of progress, mechanistic interpretability still has considerable distance to cover before achieving satisfactory progress toward most of its scientific and engineering goals.

To achieve these goals, the field not only needs greater application of current state-of-the-art mechanistic interpretability methods, but also requires the development of improved techniques. The first major sec-

tion (Section 2) therefore discusses open problems related to the methods and foundations of mechanistic interpretability (Figure 1).

We then explore key axes of research progress that will determine how far we can advance toward the goals of mechanistic interpretability (Section 3.1). Section 3 outlines how applications of interpretability methods have made progress toward the field's goals, and, for each goal, discusses the specific axes along which progress is likely needed to achieve specific objectives.

Finally, we note that the goals, applications, and methods of mechanistic interpretability do not exist in a vacuum. Like any scientific field, they lie within a broader societal context. The final section of this review examines open socio-technical problems in mechanistic interpretability (Section 4). It discusses current initiatives and possible pathways to translate technical progress into levers for AI governance, alongside consequential social and philosophical challenges faced by the field.

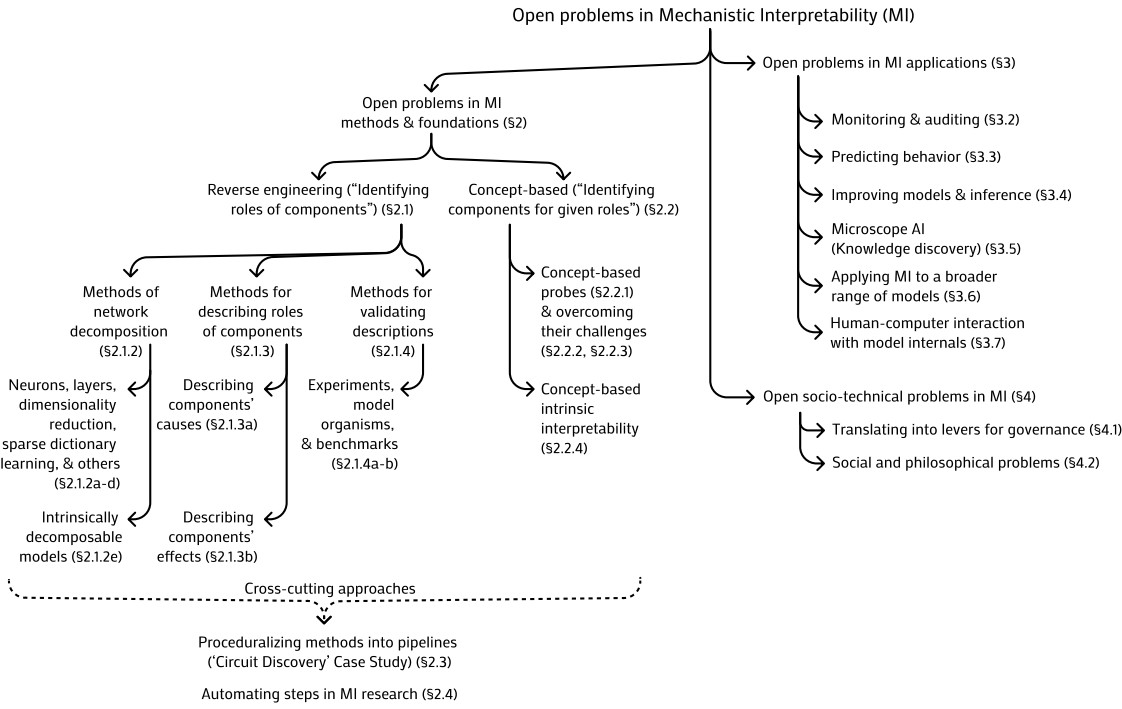

Figure 1: Summary of the core sections of this review.

## 2 Open problems in mechanistic interpretability methods and foundations

One way of thinking about what neural networks do internally is that they learn parameters that implement neural algorithms. These neural algorithms take data as input and, through a series of steps, transform their internal activations to produce an output. Different parts of the network learn different steps in the algorithm; mechanistic interpretability aims to describe the function of different network components.

The first approach, often called 'reverse engineering', is to decompose the network into components and then attempt to identify the function of those components (Section 2.1). This approach "*identifies the roles of network components*". This approach involves three distinct methods for three key steps: first, methods for neural network decomposition to break down the network into simpler parts (Section 2.1.2); second, methods for description of components to formulate hypotheses about their functional roles and interactions, such as max-activating dataset examples, attribution methods, feature synthesis methods, or causal intervention methods (Section 2.1.3); and third, validation of these descriptions to test the correctness of the hypotheses, for example through testing descriptions' predictions, attempting to use the descriptions for downstream tasks, or use of benchmarks (Section 2.1.4).

Conversely, the second approach, sometimes referred to as 'concept-based interpretability', proposes a set of (human-derived) concepts that might be used by the network and then looks for components that appear to correspond to those concepts (Section 2.2). This approach thus "*identifies network components for given roles*". Variants on this approach have been developed in order to overcome some of its issues, such as the issue that the approach often needs carefully chosen data for well-defined concepts (Section 2.2.2) or the issue that simple variants of the method only detect correlational, rather than causal variables (Section 2.2.3). Related methods build human concepts into the training of models such that they are more interpretable by default (Section 2.2.4).

In this section, we will examine both approaches ('Reverse engineering' and 'Concept-based interpretability') (Figure 2), discussing their methods and open problems. We will also touch on open problems that cut across either approach, including proceduralizing the mechanistic interpretability pipeline (Section 2.3) and its uses in automating interpretability research (Section 2.4). We provide a high-level overview in Table 1, summarizing the principal methodologies that have been used in mechanistic interpretability.

### 2.1 Reverse engineering: Identifying the roles of network components

### 2.1.1 Reverse engineering is necessary because AI and humans use different representations and perform different tasks

Large language models produce text that closely resembles human writing, and it is tempting to assume that they generate it through cognitive processes similar to those of human writers[1]. However, humans and AI models often solve problems in different ways. For example, a model that is only 1% the size of GPT-3 outperforms humans on next-token prediction tasks (Shlegeris et al., 2024). Inversely, even state-of-the-art multimodal LLMs struggle with tasks that a four-year-old could easily master, such as learning causal properties of new objects involving simple lights and shapes (Kosoy et al., 2023). An even clearer sign that humans and AI are using different representational processes are cases where humans cannot solve a problem at all, as in the case of predicting a protein structure from sequence (Jumper et al., 2021; Lin et al., 2023).

Even when both humans and AI exhibit comparable levels of competence on a given task, they may use different heuristics. For example, research shows that image models tend to rely more heavily on textural features (such as recognizing elephants by their hide rather than their shape (Geirhos et al., 2019)) or rely on dataset correlations (as when identifying fish by the fingers that proud fisherman use to hold them (Brendel & Bethge, 2019)) to a greater degree than people do. Even simple algorithmic tasks, like modular addition, which humans might solve with simple carries, were solved by a small transformer model by learning a Fourier transform strategy that researchers only understood in retrospect (Nanda et al., 2023a).

---

[1]However, even if this assumption were true, understanding LLMs would remain challenging, as we also lack a mechanistic understanding of the cognitive processes involved in human text creation!

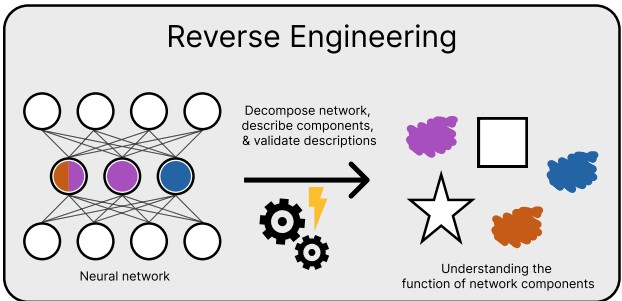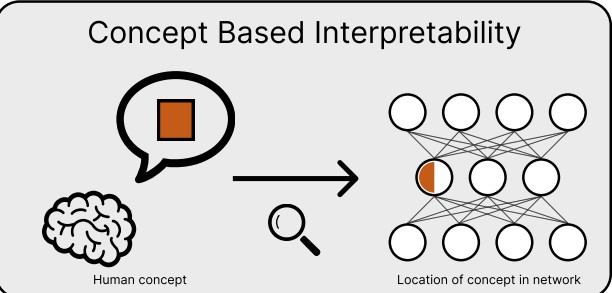

Figure 2: Two approaches to neural network interpretability. (Left) Reverse Engineering is characterized by decomposing networks into functional components and describing how those components interact to produce the network's behavior. It thus aims to 'identify the roles of network components' (Section 2.1). (Right) Concept-based interpretability on the other hand attempts to discover human concepts within neural network internals. It thus aims to 'identify the network components for given roles' (Section 2.2).

To grasp the potentially alien cognition of these models, we must develop methods to uncover and understand the previously unknown concepts and mechanisms implemented within them. In other words, we must be able to reverse engineer these models (Olah, 2024).

Reverse engineering generally involves three steps, whether it is an engine, a piece of software, or a neural network (Figure 3):

1. **Decomposition**: Breaking down the object of study into simpler parts (Section 2.1.2);

2. **Description of components**: Formulating hypotheses about the functional role of component parts and how they interact (a process that can be called 'interpretation') (Section 2.1.3);

3. **Validation of descriptions**: Testing if our hypotheses are correct (Section 2.1.4).

If our hypotheses are invalidated, we must either improve our decomposition or improve our hypotheses regarding the functional roles of its components. We will systematically examine each step, analyzing current methods and their respective shortcomings.

### 2.1.2 Reverse engineering step 1: Neural network decomposition

In mechanistic interpretability, our aim is to decompose a neural network and study its parts in isolation in order to explain how neural networks generalize. This aim raises the question of how best to carve a neural network "at its joints" for the purposes of interpretability.

**Networks do not naturally decompose into architectural components.** The naive approach to decompose neural networks involves breaking them down into their architectural components, such as individual neurons, attention heads, or layers.

Some early attempts to interpret deep artificial neural networks studied the responses or weight structure of individual neurons or single convolutional filters (Erhan et al., 2009; Le et al., 2012; Krizhevsky et al., 2012; Szegedy et al., 2014; Simonyan et al., 2014b; Zhou et al., 2015; Srivastava et al., 2014; Yosinski et al., 2015; Mordvintsev, 2015; Olah et al., 2017a; Bau et al., 2020; Dalvi et al., 2019; Olah et al., 2020a; Cammarata et al., 2020). These efforts paid homage to the 'Neuron Doctrine' in neuroscience, which posits that individual neurons are the structural and functional unit of the nervous system (Cajal, 1924; Sherrington, 1906). However, researchers discovered that single neurons are 'polysemantic' – they seem to respond to multiple kinds of features in both artificial (Wei et al., 2015; Nguyen et al., 2016c; Olah et al., 2017a) and biological networks (Churchland & Shenoy, 2007; Rigotti et al., 2013; Mante et al., 2013; Raposo et al., 2014). These observations support earlier theoretical work that suggested representations used by neural networks do not necessarily align with the activation of individual neurons (Hinton, 1981).

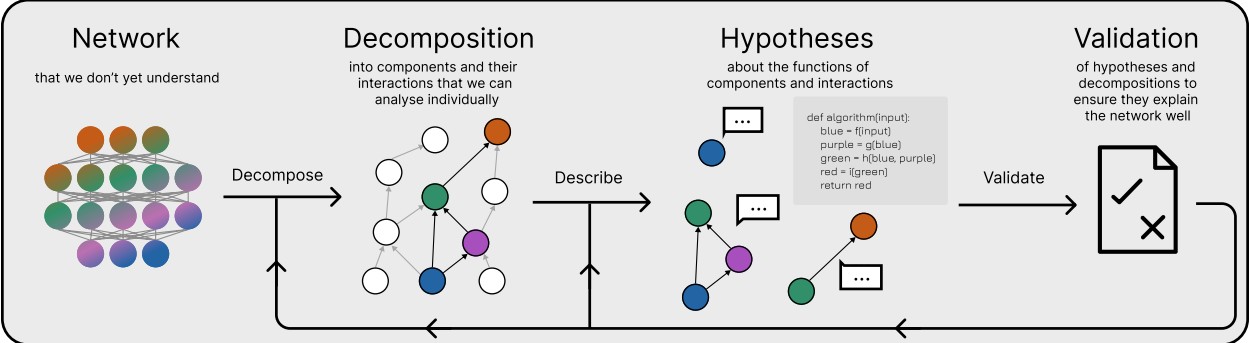

Figure 3: The steps of reverse engineering neural networks. (1) Decomposing a network into simpler components. This decomposition might not necessarily use architecturally-defined bases, such as individual neurons or layers (Section 2.1.2). (2) Hypothesizing about the functional roles of some or all components (Section 2.1.3). (3) Validating whether our hypotheses are correct, creating a cycle in which we iteratively refine our decompositions and hypotheses to improve our understanding of the network (Section 2.1.4).

Interpreting individual attention heads does not fare better than interpreting individual neurons, as attention heads also exhibit polysemanticity (Janiak et al., 2023). More broadly, research suggests that studying the attention patterns of models can often be misleading (Jain & Wallace, 2019; Pruthi et al., 2020).

Some work even suggests that representations in language models might span multiple layers (Yun et al., 2021; Lindsey et al., 2024; Ameisen et al., 2025). This chimes with work that edits or intervenes on individual layers, which indicates that this level is too coarse-grained to robustly carve the network at its joints (Meng et al., 2022b; Wang et al., 2023).

If natural architectural components, such as individual neurons, attention heads, or layers, do not provide a natural way to decompose neural network representations, then what does?

**Decomposition by dimensionality reduction methods.** If individual neurons are not the right decomposition, perhaps groups or patterns of neurons are. Many decomposition methods attempt to identify activation vectors that correspond to the basic unit of neural network computation. One common approach is to provide models with a range of unlabeled inputs, collect the resulting hidden activations, and then apply unsupervised dimensionality reduction techniques to these hidden activations. The hope is that structure in the hidden activations corresponds to the structure of neural computation. Commonly used dimensionality reduction methods include Principal Component Analysis or Singular Value Decomposition (Hollinsworth et al., 2024; Marks & Tegmark, 2024; Huang et al., 2024a; Bushnaq et al., 2024), tensor factorization (Oldfield et al., 2023), and non-negative matrix factorization (Olah et al., 2018; Voss et al., 2021; Cammarata et al., 2020; Fel et al., 2023), though these techniques are no longer predominant methods used for mechanistically decomposing language models (Friedman et al., 2024). Related approaches use singular value decomposition as an initial step, followed by unsupervised clustering step in order to identify semantically unique and distinguishable concepts (Graziani et al., 2023).

**Decomposition by sparse dictionary learning (SDL).** According to the 'superposition hypothesis', neural networks are capable of representing more features than they have dimensions, as long as each feature activates sparsely (Elhage et al., 2021) (Figure 4). This is a key reason that dimensionality reduction methods are not considered state-of-the-art, because they cannot identify more directions than there are activation dimensions. The superposition hypothesis, coupled with the failure of dimensionality reduction methods to overcome it, motivated the search for methods that can identify more directions than dimensions. Recent work has explored the use of sparse dictionary learning (SDL) to this end (Elhage et al., 2021; Sharkey et al., 2022b; Huben et al., 2024; Bricken, 2023).

Currently, SDL is the most popular set of unsupervised decomposition methods in mechanistic interpretability. SDL encapsulates a family of methods, including Sparse Autoencoders (SAEs) (Gao et al., 2024; Tem-

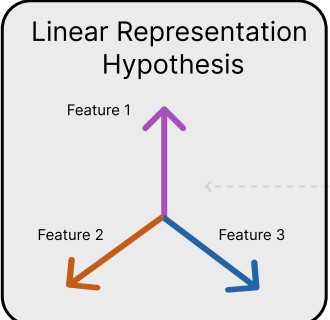 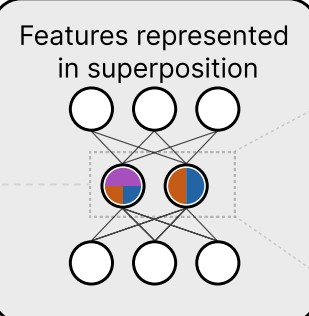 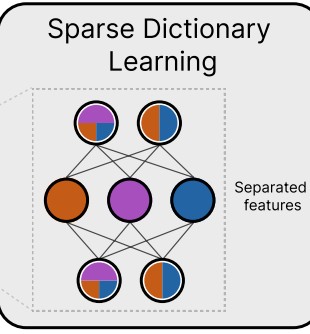

Figure 4: Three ideas underlying the sparse dictionary learning (SDL) paradigm in mechanistic interpretability. (Left) The linear representation hypothesis states that the map from 'concepts' to neural activations is linear. (Middle) Superposition is the hypothesis that models represent many more concepts than they have dimensions by representing them both sparsely and linearly in activation spaces. (Right) SDL attempts to recover an overcomplete basis of concepts represented in superposition in activation space.

pleton et al., 2024; Rajamanoharan et al., 2024; Makelov et al., 2024; Kissane et al., 2024b; Braun et al., 2024), Transcoders (Dunefsky et al., 2024; Ameisen et al., 2025), and Crosscoders (Lindsey et al., 2024; Ameisen et al., 2025).

In SDL, hidden activations are typically passed to a small neural network consisting of only two layers, which correspond to an encoder and decoder respectively, with a wide hidden space. The encoder activations represent how active each 'latent' [2] is, and the decoder matrix corresponds to a dictionary of latent directions. We want to train the dictionary elements to align with 'feature directions' in the network's hidden activations. Since we assume that individual features are sparsely present in the activations, the encoder activations are trained to be sparsely activating. In the case of SAEs, the output is trained to either reconstruct the input or, in the case of transcoders (Dunefsky et al., 2024; Ameisen et al., 2025), to reconstruct the activations of the next layer or layers. Crosscoders (Lindsey et al., 2024; Ameisen et al., 2025) permit a wider class of inputs and outputs, potentially reconstructing the activations of many layers simultaneously. Since the encoder is nonlinear, it is thought to be able to learn to activate a latent only if a feature is 'active' in the hidden activations and remain 'off' if it is not.

Although SDL is considered a leading decomposition method for mechanistic interpretability, it has substantial practical and conceptual limitations (Figure 5).

**SDL reconstruction errors are too high**: Large errors in SDL reconstruction raise the question whether SDL methods can reconstruct the hidden representation sufficiently well such that the latents learned by SDL are adequately faithful to the models being interpreted. To measure this, the model's true hidden activations can be replaced with sparse dictionary reconstructions, then subsequently evaluating the extent to which the model's performance decreases. In practice, the error results in significant performance reductions. When a sparse dictionary with 16 million latents was inserted into GPT-4, the language modeling loss was equivalent to a model with only 10% of GPT-4's pretraining compute (Gao et al., 2024). Similarly, Makelov et al. (2024) found that using reconstructions from sparse autoencoders decreased GPT-2 small performance by 10% when trained on task-specific data, and 40% when trained on the full distribution.

Making sparse dictionaries much larger and sparser to reduce errors is a feasible but computationally expensive approach. Furthermore, in the limit, this results in merely assigning one dictionary latent per datapoint, which is clearly less interpretable. One partial solution is using SDL methods with 'error nodes' (Marks et al., 2024), to account for the discrepancies between the original and reconstructed activations. However, while the sparse autoencoders identified interpretable latents, the error nodes contain 'everything else', making them an inadequate solution to the problem. Engels et al. (2024b) found that these reconstruction errors are not purely random, as much of the direction of the error and its norm can be linearly predicted from

---

[2]The term latent is often used instead of the word feature to refer to SDL dictionary elements, since the term 'feature' is often used to refer to multiple different ideas (Smith, 2024b)

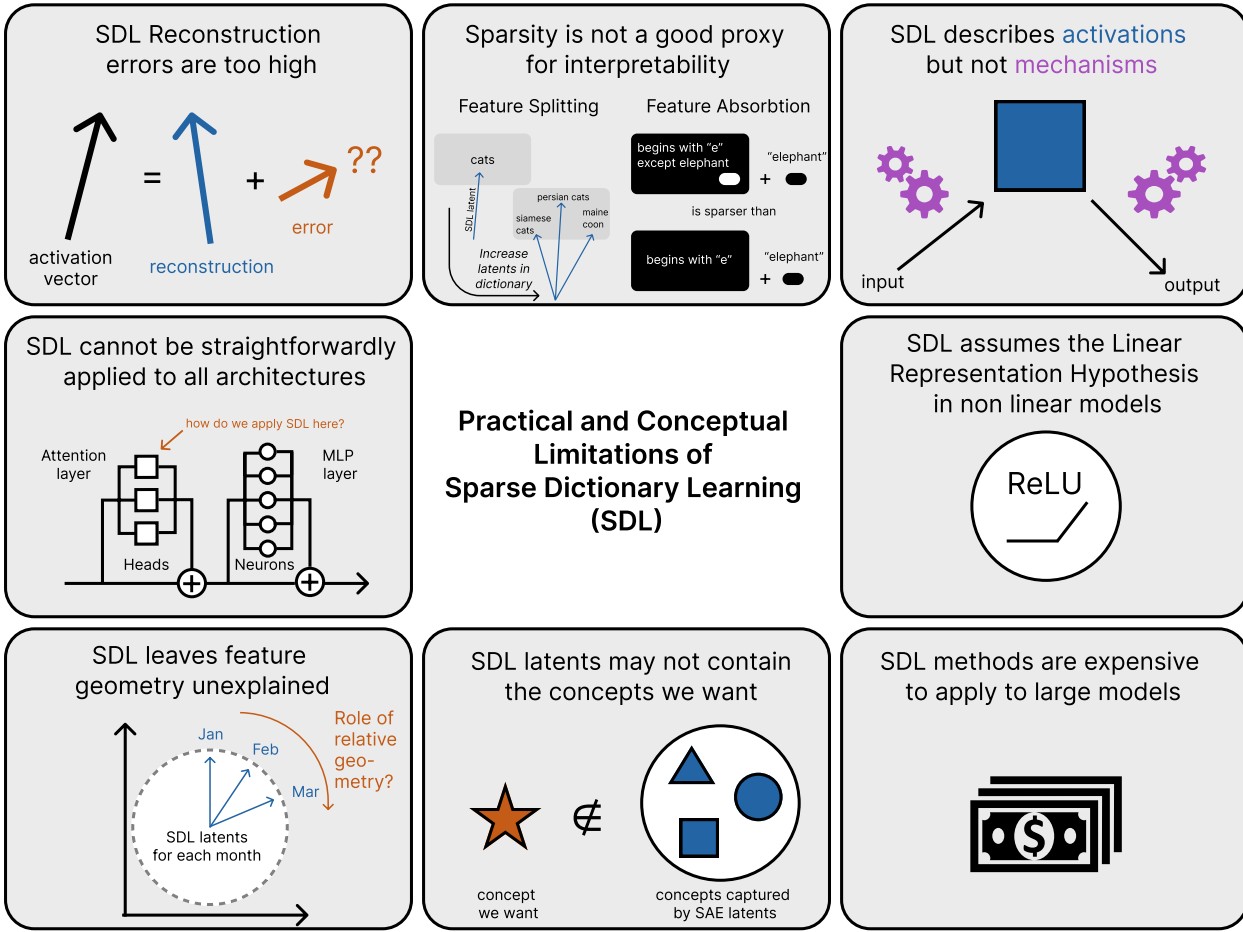

Figure 5: Sparse dictionary learning has a number of practical and conceptual limitations that cause issues when using it to reverse engineer neural networks (Section 2.1.2).

the initial activation vector. This suggests that current SDL methods may systematically fail to capture certain structured aspects of model representations, but also implies potential solutions. To overcome this issue, it may be necessary to improve SDL training methods, or develop entirely new methods of network decomposition.

**SDL methods are expensive to apply to large models**: SDL involves training a small neural network for every layer of the AI model that we want to interpret. Typically, the sparse dictionaries have more parameters at that layer than the original model does, and consequently will probably be relatively expensive to train compared to the original model. [3] As AI models become larger, scaling costs of SDL also increase, although it remains unclear whether relative scaling costs are sub- or supra-linear. For cost effectiveness, it may be important to develop intrinsically decomposable methods for training models, while remaining at (or close to) state-of-the-art performance (Section 2.1.2). This approach will help avoid incurring the expense of both training and decomposing AI models.

**SDL assumes the linear representation hypothesis in nonlinear models**: Other problems with SDL arise from the assumptions on which SDL is based. One such assumption is the linear representation hypothesis. The linear representation hypothesis observes that though neural networks are nonlinear functions and could potentially use highly nonlinear representations (Black et al., 2022; Engels et al., 2024b; Kirch et al.,

---

[3]The actual relative cost is unclear since there are no public attempts to apply SDL to every vector space in a model, although some work applies SDL to various layers (Marks et al., 2024; Gao et al., 2024; Braun et al., 2024; Lieberum et al., 2024; Bloom & Lin, 2024a; Huben et al., 2024)

2024), they tend to use representations that exhibit strikingly linear behavior (Smolensky, 1986; Mikolov et al., 2013; Olah et al., 2020b; Elhage et al., 2021; Park et al., 2023a; Guerner et al., 2024; Gurnee & Tegmark, 2024). The hypothesis formalizes this phenomenon by stating that high level concepts are linearly represented as directions (vectors) in neural network embeddings. Its two core claims are (i) that the composition of multiple concepts can be represented as the addition of their corresponding feature vectors, and (ii) that the intensity of a concept is represented by the scale of its corresponding feature vector (Olah & Jermyn, 2024). Earlier work (Elhage et al., 2021) defined the linear representation hypothesis with the additional assumption of one-dimensional features, but this definition has since been refuted (Yedidia, 2023; Chughtai & Lau, 2024; Engels et al., 2024a) and clarified (Olah & Jermyn, 2024). Another recently proposed criterion is that the intensity of concepts should be retrievable by a linear function of the embeddings, up to a small error (Hänni et al., 2024; Olah & Jermyn, 2024).

A weak version of the linear representation hypothesis states that some concepts are linearly represented, while a strong version may assert that all concepts are (Smith, 2024a). Some works have shown that the strong version is false for some models (Black et al., 2022; Csordás et al., 2024). The weaker version of the linear representation hypothesis is supported by the successes of linear probes (Section 2.2.1), activation steering (Section 3.2.2), and the success of sparse autoencoders in finding seemingly interpretable latents (Section 2.1.2).

**Sparsity is not a good proxy for interpretability**: Another of SDL's key assumptions is that feature activations are sparse. Therefore, SDL methods optimize their latents accordingly to be sparsely activating, with the implicit assumption that sparser decompositions are more interpretable than denser ones. However, this assumption may not necessarily hold. The (related) problems of feature splitting (Bricken, 2023), feature absorption (Chanin et al., 2024) and composition (Till, 2024) suggest that with sufficient optimization pressure, sparsity as a proxy for interpretability breaks down (though note that it is debated whether feature splitting is actually a problem). Other proxies, such as minimum description length, might be better optimization targets than sparsity *per se* (Ayonrinde et al., 2024). It also may be the case that no proxy metric is sufficient.

**SDL leaves feature geometry unexplained**: SDL decomposes networks into single directions in the activation space, which is a reasonable approach only if we consider feature activations akin to 'a bag of features' (Harris, 1954) without any internal structure. However, the geometric arrangement of features in relation to each other seems to reflect semantic and functional structure (Engels et al., 2024a; Gurnee & Tegmark, 2024; Park et al., 2024b; Bussman et al., 2024). Understanding the geometric structure of a network means understanding the position of one feature relative to (potentially) many others. Comprehending this may be necessary if we want to know why and how a network treats certain features similarly or differently. If understanding the global geometry (all-to-all relationships) of features is essential to understand neural networks, this might pose a fundamental problem for current approaches to mechanistic interpretability (Hänni et al., 2024; Mendel, 2024). However, if only local geometric relationships between features need to be understood, understanding networks with a 'bag of features' approach may be more feasible.

**SDL cannot straightforwardly be applied to all architectures**: SDL was originally developed to identify features that may be represented in superposition across many neurons within a single layer. However, representations may be spread over other network architecture components besides neurons. In transformers, for example, representations may be spread across separate attention heads (Jermyn et al., 2023; Janiak et al., 2023), and even across different layers (Yun et al., 2021; Lindsey et al., 2024). However, it is not immediately obvious how to decompose representations distributed across attention heads with SDL (Mathwin et al., 2024; Wynroe & Sharkey, 2024). Nor is it straightforward to translate cross-layer distributed representations into causal descriptions of neural network mechanisms (Lindsey et al., 2024), though some recent work makes progress toward this (Ameisen et al., 2025).

**SDL decomposes the input and output activations of network mechanisms, but not the mechanisms themselves**: Ultimately, the aim of mechanistic interpretability is to understand the mechanisms learned by neural networks. The parameters of the network, along with its nonlinearities and other architectural components, implement these mechanisms, which are applied to the input's hidden activations. SDL identifies directions in activation space. Being activations, they only interact with the network's mechanisms,

but are not the mechanisms themselves. Describing the mechanisms directly remains unresolved with SDL. Gaining insights about the network's mechanisms from SDL latents requires further post hoc analysis, which can be labor intensive, computationally expensive, or data set dependent (Huben et al., 2024; Bricken, 2023; Riggs et al., 2023; Marks et al., 2024). This is an instance of a more broad problem with current mechanistic interpretability work; we primarily focus on understanding neural network activations, with little attention paid to how this structure in activations is computed via weights (Chughtai & Bushnaq, 2025).

**SDL latents may not contain the concepts needed for downstream use cases**: When using SAEs for practical tasks, there sometimes exists a single latent or small set of latents representing some concept of interest for task performance (and not representing much else). For instance, Kantamneni et al. (2024) found a single latent whose activation pattern was more accurate than official dataset labels on the NLP task GLUE CoLA (Warstadt et al., 2019). More often than not though, a sparse set of latents that encode some useful concept of interest do not exist. It is unclear what causes this problem. One hypothesis is that the concept we want isn't how the model 'thinks' about the concept, and the SAE is working as intended. Alternatively, the SAE training distribution could be too narrow, resulting in the SAE not being incentivised to learn the important latents. (Kissane et al., 2024c) found that SAEs trained on pretraining data generally do not have good latents for the concept of 'refusing' harmful user requests, while SAEs trained on chat formatted data do. Or, the SAE might not have a large enough dictionary size to learn all concepts of interest. Many more hypotheses are plausible. A complicating factor in using SDL to identify the learned mechanisms of neural networks is that the latents identified by SDL depend heavily on the data set used to train them. This is an undesirable property for a decomposition method that was initially hoped to be capable of identifying the fundamental units of computation in neural networks (Kissane et al., 2024c).

**Current decomposition methods lack solid theoretical foundations.** Given the practical and conceptual issues with SDL, there is broad agreement that the question of how to correctly decompose networks into atomic units remains a central problem, evidenced by the large amount of effort focused on the direction in recent years. After investing considerable effort in SDL approaches, it is apparent that improved conceptual clarity beyond the idea of superposition (Elhage et al., 2021) is needed to advance neural network decomposition.

One of the most significant open questions is the absence of clarity around the nature of features, despite being the central focus of SDL's identification efforts. Satisfying formal definitions are elusive and conceptual foundations are not yet established. However, even without foundations, progress in mechanistic interpretability is possible — even confused concepts can be pragmatically useful (Henighan, 2024).

Without solid conceptual foundations, it remains unclear whether the superposition hypothesis, which underpins the SDL paradigm, is fundamentally valid or merely pragmatically useful (Henighan, 2024; Templeton et al., 2024). If it is the latter, there may be better methods than SDL for carving neural networks at their joints. Such methods may take into account feature geometry or take a more dynamic, developmental view of how mechanistic structure emerges in the training process (Hoogland et al., 2024; Wang et al., 2024b). Moreover, although there is much emphasis on how models represent features in superposition, there is comparatively less work examining how they might perform computation on them natively in superposition. Further conceptual work into this problem may suggest new methodologies for decomposing networks, or provide bounds on the number of features we should expect models to be capable of learning (Hänni et al., 2024; Adler & Shavit, 2024; Bushnaq & Mendel, 2024). The field should explore very different approaches that show promise for decomposing neural networks into functional components and that do not exhibit many of the problem of the SDL paradigm, such as decomposing or analyzing network parameters (Braun et al., 2025; Tsang et al., 2018) or methods that decompose networks according to analytical descriptions of the local functions they implement, such as deep Taylor decomposition (Montavon et al., 2017).

Mechanistic interpretability should ideally be built on more formal foundations. Some work attempts to ground mechanistic interpretability in the formalisms of causality (Geiger et al., 2024a). However, this approach has not yet yielded canonical causal mediators (Mueller et al., 2024) that may form a basis for decomposition methods. How should we go about finding them? Given that our field's objective is to understand the learned structures that underlie networks' generalization behaviors, exploring theories about why neural networks generalize appears to be a promising avenue. But theories that attempt to characterize

why neural networks generalize, such as the spline theory of neural networks (Balestriero & richard baraniuk, 2018), theories of neural networks' simplicity bias (Valle-Perez et al., 2019), deep learning theory involving the neural tangent kernel (Jacot et al., 2018; Roberts et al., 2022), or singular learning theory (Watanabe, 2009; Wei et al., 2023) have either not yet yielded mathematical objects that can be easily used for interpretability, or simply have not been successfully linked to approaches for interpreting neural networks. Establishing these connections would make significant progress toward carving neural networks at their joints to facilitate mechanistic interpretability. And if we can carve trained networks at their joints, it may suggest ways to train networks such that they come 'pre-carved'. Thus, better theoretical foundations may also be important for developing models that are intrinsically decomposable by design, which we discuss next.

**Intrinsic interpretability: Building more easily decomposable models** The current strategy of training a model solely for performance and then interpreting it post hoc may not be optimal if our goal is a model that is both interpretable and performant. To this end, it may be beneficial to prioritize interpretability during model training, for which there are several plausible approaches.

Instead of post-hoc decomposing trained network activations into discrete codes (Section 2.1.2), network activations could be forced to use a discrete code from the outset, as in Tamkin et al. (2025). MLPs could also be trained with sparser activation functions, such as TopK (Makhzani & Frey, 2013; Bills et al., 2023) or SoLU (Elhage et al., 2022). Similar approaches could be potentially used to restrict attention superposition (Jermyn et al., 2023) by limiting the number of heads attending to any query-key pair. Several approaches, such as 'mixture of experts' (Shazeer et al., 2017; Fedus et al., 2022; He, 2024), use sparsely activating components — with a large enough number of experts, and with sufficient activation sparsity, experts may become individually interpretable (Warstadt et al., 2019). Many attempts to incentivize interpretable activations directly so far have not been competitively performant, and have also allowed 'superposition to sneak through', mitigating benefits.

We can also target weight sparsity directly during training, including approaches such as $L_0$ regularization (Louizos et al., 2018) or pruning (Mozer & Smolensky, 1988; Frankle & Carbin, 2019; Mocanu et al., 2018). Han et al. (2015) achieve sparse weights by using magnitude pruning followed by finetuning. This highlights a general strategy of finetuning with interpretability in mind (also used by Tamkin et al. (2025)), avoiding the potentially excessive cost of training from scratch. Another related strategy is targeting modularity (Kirsch et al., 2018; Andreas et al., 2016). For example, brain-inspired modular training (Liu et al., 2023) trains for modularity by embedding neurons in a geometric space and encouraging geometrically local connections.

Existing research implements other approaches that can simplify linearity-reliant circuit analysis (Elhage et al., 2021). For example, there is work removing the layer norm operations (Heimersheim, 2024), using input-switched affine transformations for recurrence (Foerster et al., 2017). Other studies leverage architectures that are mathematically analyzable in other ways than linearity, such bilinear activations in MLPs (Sharkey, 2023; Pearce et al., 2024).

After decomposing a network into components, whether through post hoc decomposition or by using intrinsically decomposable models, our task remains unfinished. We must provide an interpretation of the functional role of each component (Step 2 – Section 2.1.3) and validate that interpretation (Step 3 – Section 2.1.4). In the next section, we will discuss common methods of interpretation and their shortcomings.

### 2.1.3 Reverse engineering step 2: Describing the functional role of components

After decomposing networks into parts, the next step of reverse engineering is to "describe" the functional role of these components. This step is best thought of as generating hypothesized 'interpretations' or 'explanations'. These explanations form candidate descriptions of the functional role of a given component, and should not be taken to be definitive conclusions before they are thoroughly validated (see Section 2.1.4).

Descriptions of the functional role of neural network components can either indicate (1) the cause of a component's activation or (2) what occurs after that component has been activated, or – preferably – both (Figure 6). In this section, we will discuss the existing set of tools available to mechanistic interpretability researchers to describe the functional role of network components.

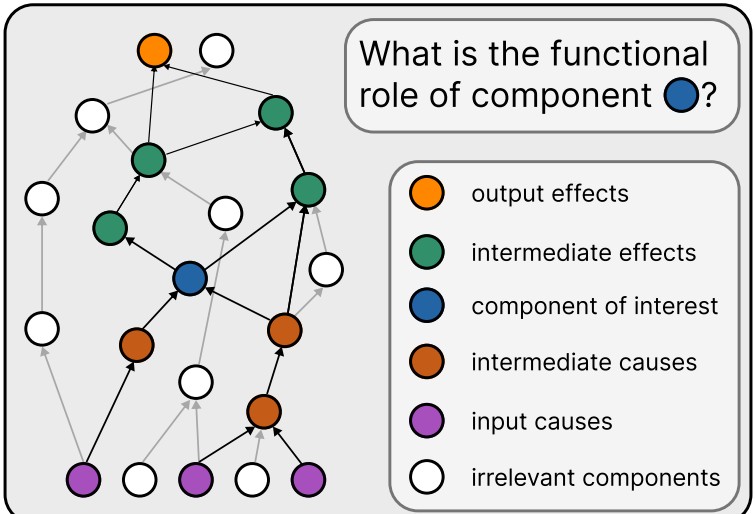

Figure 6: To study the functional role of the blue component numerous approaches are possible. We could study its causes: the purple input components or the red intermediate components via e.g. feature synthesis (Olah et al., 2017a; 2020b), maximum activating examples (Olah et al., 2017a; Bricken, 2023), or attributions (Sundararajan et al., 2017). Or we could study its effects: the orange output components or the green intermediate components via e.g. the logit lens (Nostalgebraist, 2020), activation steering (Turner et al., 2024), or attributions (Marks et al., 2024).

**Explanations for what causes components to activate**  Explanations of the causes of component activation can use three broad categories of methods, each with several problems: (1) Highly activating data set examples; (2) Attribution methods; and (3) Feature synthesis.

**Highly activating data set examples**. The simplest method is to use highly activating data set examples (sometimes called 'exemplar representations' (Hernandez et al., 2022)). These are inputs on which a particular component is strongly activated. For a given component, analyzing apparent commonalities in the inputs suggests hypotheses for what causes that component to activate. This step may be carried out by humans or AI systems (Section 2.4).

Despite being widely used, this approach has several substantial issues. The first issue is that the method relies on human prior beliefs, which may lead interpreters to project their human understanding onto models that may, in fact, be using unfamiliar concepts. This bias could lead us to identify concepts in the model that do not truly explain the model's functioning (Freiesleben & König, 2023; Donnelly & Roegiest, 2019; Gale et al., 2020).

Another issue with this approach is the potential for 'interpretability illusions'. Bolukbasi et al. (2021) show that bias in data sets can create misleading explanations even when top activating examples are selected from real data sets, as opposed to synthetically created data. Depending on the data set from which the examples were drawn, human annotators identified dramatically different meanings for given directions in the activation space of BERT (Devlin, 2018).

A third issue is that this approach often yields plausible explanations for arbitrarily chosen directions in the activation space (Szegedy et al., 2014). This means that plausible explanations based on highly activating data set examples cannot be solely relied upon to identify the basic units of computations in neural networks — other methods are needed to accurately identify them. Furthermore, it is possible to develop adversarial models that deliberately yield misleading feature visualizations (Geirhos et al., 2025).

Many of the issues with highly activating data set examples stem from the fact that they merely provide correlational explanations for the activation of a network component, rather than causal explanations. To identify causal explanations, other methods, such as attribution methods, are necessary.

**Attribution methods are necessary for causal explanations but are often difficult to interpret**. Attribution methods (Baehrens et al., 2009; Simonyan et al., 2014a; Nguyen et al., 2016b; Selvaraju et al., 2019; Sundararajan et al., 2017; Fong & Vedaldi, 2017; Lundberg & Lee, 2017; Ribeiro et al., 2016) are intended to measure the causal importance of upstream variables (such as inputs) on downstream variables (such as a network component).

Specific examples of attribution methods include Grad-CAM (Selvaraju et al., 2019) and Integrated Gradients (Sundararajan et al., 2017), as well as perturbation or model-agnostic approaches such as LIME (Ribeiro et al., 2016) and SHAP (Lundberg & Lee, 2017).

Attributions methods are often categorized as gradient-based (Mozer & Smolensky, 1988; Baehrens et al., 2009; Simonyan et al., 2014a; Nguyen et al., 2016b; Selvaraju et al., 2019; Sundararajan et al., 2017; Wang et al., 2024d) or sampling-, perturbation-, or ablation-based (Fong & Vedaldi, 2017; Vig et al., 2020; Geiger et al., 2020; Ghorbani & Zou, 2020; Meng et al., 2022b; Chan et al., 2022a; Nanda, 2023a). However, on a theoretical level, many gradient-based methods identify only a first-order approximation of the ideal attribution, which is sometimes a poor approximation (Watson, 2022). Most approaches only attribute important single components, though some work, such as integrated Hessians, attributes important feature interactions (Janizek et al., 2020). Further extending the understanding of feature interactions, various methods have been proposed. For instance, KernelSHAP-IQ characterizes higher-order Shapley Interaction Index as a solution to a weighted least square problem, offering one way to quantify such interactions (Fumagalli et al., 2024). Specific to Graph Neural Networks (GNNs), recent advancements such as GraphSHAP-IQ by Muschalik et al. (2025) enable the exact and efficient computation of any-order Shapley Interactions. Other specialized methods, like BiLRP, have been developed to identify feature interactions in specific contexts such as text similarity models (Vasileiou & Eberle, 2024). Adebayo et al. (2018) revealed more practical implications, demonstrating that some gradient-based methods identify attributions that are independent both of the model and of the data generating process. Furthermore, an adversary can train a model or perturb an input to reveal any attribution map (Dombrowski et al., 2019; Ghorbani et al., 2019; Heo et al., 2019; Kindermans et al., 2019; Slack et al., 2020; Zhang et al., 2020)[4]. For specific architectures like Graph Neural Networks (GNNs), methods such as GNN-LRP have been developed to provide higher-order explanations by identifying relevant walks in the input graph using Layer-wise Relevance Propagation (Schnake et al., 2022). Perturbation methods present certain issues, such as taking models off their training distribution and eliciting unusual behavior, among other theoretical complications (Feng et al., 2018; Molnar et al., 2021; Hooker et al., 2021; Molnar et al., 2024; Freiesleben & König, 2023; Slack et al., 2021). Developing efficient and accurate attribution methods thus remains an open problem.

**Feature synthesis**. Feature synthesis is a strategy that combines highly activating data set examples and gradient-based attribution methods (Erhan et al., 2009; Szegedy et al., 2014; Olah et al., 2017a). This approach attempts to synthesize inputs that maximize the activation of a component subject to some regularization, such as consistency with a generative model (Nguyen et al., 2016a; 2017) or total variation distance (Mahendran & Vedaldi, 2014). However, criticisms of feature synthesis methods suggest that natural dataset examples may serve interpretation better (Zimmermann et al., 2021; Borowski et al., 2021) or show that current methods struggle to identify trojans (Casper et al., 2023a).

**Explanations for the downstream effects of components**  Alternatively, the functional role of a component can be described through its downstream effects.

**Studying the direct effect**. The logit lens (Nostalgebraist, 2020) applies the models unembedding matrix to an intermediate residual stream representations, converting it into a distribution over the model's output vocabulary. Direct logit attribution is a generalization of this technique, applying to any appropriately sized vector space in the model, e.g. the output of MLP layers (Geva et al., 2021; 2022b;a; Dar et al., 2023), attention blocks, gradients (Katz et al., 2024) of these, and SDL decoder weights (Bricken, 2023). Adding a trainable affine or linear transformation before unembedding, as also referred to as the tuned lens, improves decoding accuracy (Belrose et al., 2023a; Yom Din et al., 2024), at the cost of less faithfully representing when the model has completed computation.

---

[4]However, Freiesleben & König (2023) argue that adversarial examples do not truly undermine saliency maps as these are highly dissimilar to real interpretability challenges.

In the language of causality (Pearl, 2009), unembedding a residual stream vector measures the direct effect of that vector on the output (McGrath et al., 2023). However, the logit lens cannot measure the indirect effect, the effects resulting from the influence that the embedding has on the hidden activations of subsequent layers. Other methods, such as causal interventions (see below), are necessary to measure the indirect effect. In the future, it may be possible to extend logit-lens-like approaches to not only project the effects of network components on directions in output vocabulary space, but also on intermediate downstream components.

**Causal Interventions**. Causal interventions typically substitute ("patch") the value of some network component, usually an activation vector, with a different value during a forward pass, observing the resulting effect on the model. There exist a related set of techniques: ablation (for the special case of zero-ing or otherwise attempting to delete activations entirely), activation patching, causal mediation analysis, causal tracing, and interchange intervention (Vig et al., 2020; Geiger et al., 2020; Meng et al., 2022a; Chan et al., 2022a; Nanda, 2023a). [5]

More surgical patches are also sometimes also made on edges between network components. This approach, known as "path" patching, allows us to isolate the effect of one particular component on only one other component, instead of on the entire rest of the network (Goldowsky-Dill et al., 2023). Causal scrubbing (Chan et al., 2022a) is a generalization of path patching that allows for testing hypotheses concerning any given connection between a set of network components.

Causal intervention methods can also generate supervision signal to identify subspaces of interest. For instance, distributed alignment search (Geiger et al., 2024c) learns a linear subspace that represents a particular concept, using data with interventions on that concept as supervision (Geiger et al., 2024c; Guerner et al., 2024; Wu et al., 2023). Other work learns masks over network components to remove irrelevant components (De Cao et al., 2020; Csordás et al., 2021; Davies et al., 2023).

A causal intervention typically requires a forward pass of the model. This may make performing one for every network component in large models, long contexts, or when using finer-grained components such as sparse autoencoder latents prohibitively expensive. Faster alternatives that work well in practice (Marks et al., 2024; Templeton et al., 2024) include gradient-based approximations to activation patching, such as attribution patching (Nanda, 2023a; Syed et al., 2024), AtP* (Kramár et al., 2024), and integrated gradients (Sundararajan et al., 2017).

**Observing the effects of components on sequential behavior**. Another way to study the effects of model components is to patch in activated components and observe their effect on model behavior. Steering is one example of this (Rimsky et al., 2024; Turner et al., 2024), where components (activation vectors) are activated in order to influence the network's behavior, often in interpretable ways. To determine the functional role of an activation vector, a related method is to have a language model itself decode the activations (Chen et al., 2025; Watkins, 2023; Ghandeharioun et al., 2024; Huang et al., 2024b; Kharlapenko et al., 2024). These approaches, known as "patchscopes", patch activations from one forward pass of a model into a different forward pass (perhaps in a different model). The context of the new forward pass is designed to elicit relevant information from the activations of the original forward pass.

A related sequential behavior based technique is simply to read the chain-of-thought (Wei et al., 2024; Kojima et al., 2024) produced by a language model's output. While this could be considered an 'interpretability technique' in the sense that it aims to explain model decisions, it does not use model internals in those explanations, at least not directly. Recent research demonstrates that chains of thought are not entirely faithful to the model's underlying decision-making process (Agarwal et al., 2024; Atanasova et al., 2023; Turpin et al., 2023; Lanham et al., 2023; Ye & Durrett, 2022). A promising future direction for interpretability research may be to incorporate model internals into chain-of-thought training, which could incentivize faithfulness. Another possibility is building monitors based on model internals to improve transparency in chain-of-thought faithfulness.

---

[5] For instance, by patching activations from the corrupt prompt "the capital of Italy is" into the clean prompt "the capital of France is", we can observe the effect on the output ("Rome" vs. "Paris"). This tells us which component values are relevant for the differing output between the two prompts, but not the information that remains consistent (e.g. the fact that the answer is a city).

### 2.1.4   Reverse engineering step 3: Validation of descriptions

Initial descriptions of network components' functional roles should be treated as hypotheses that first require validation to ensure that they are reasonable.

Conflating hypotheses with conclusions has regrettably been commonplace in mechanistic interpretability research, making validation an important area for the field to improve (Madsen et al., 2024; Stander et al., 2025). Unfortunately, it is often hard to distinguish faithful explanations of neural network components from merely plausible ones. Numerous examples of model interpretations fail sanity checks (Adebayo et al., 2018; Leavitt & Morcos, 2020; Miller et al., 2024); "interpretability illusions" in which seemingly convincing interpretations of a model later turned out to be false (Bolukbasi et al., 2021; Makelov et al., 2023); or instances where different approaches to explaining the same phenomenon yielded different interpretations (e.g. Chan et al. (2022b) or Chughtai et al. (2023) vs. Stander et al. (2025) vs. Wu et al. (2024a)). Several other instances were previously cited in this review (Section 2.1.3). Hypotheses in interpretability require extensive validation beyond what appearances might imply.

Validating a hypothesis involves posing a simple question: Does the hypothesis make good predictions about the neural network's behavior? Testing the hypothesis often requires multiple approaches (Mueller et al., 2024). To validate descriptions, many approaches simply apply a different description method than the one used to generate the initial hypothesis (Section 2.1.3). If one description method yields a different description from another, it invalidates the hypothesis, which necessitates returning to an earlier step in the reverse engineering cycle (Figure 3). However, methods for validating descriptions are not limited to other component description methods. Hypothesis validation may take many forms, including the following:

**Predicting activations and counterfactuals**: By using natural language explanations of a given network component's function, it is possible to predict the component's activation levels on different inputs. This analysis can be carried out by humans or by AI systems (Section 2.4), as in (Hernandez et al., 2022; Bills et al., 2023; Shaham et al., 2025; Juang et al., 2024). In essence, our interpretations should enable us to successfully predict counterfactual scenarios in neural networks. For instance, if we ablate or activate particular network components, we should be able to predict specific downstream effects on other components.

**Predicting and explaining unusual failures or adversarial examples**: Good explanations of neural network behavior should help us identify and explain cases where that behavior fails to produce expected outcomes. For instance, Hilton et al. (2020) validated their methods by explaining cases where a deep reinforcement learning agent's neural network failed to achieve maximum reward and also explained specific hallucinations exhibited by the network. Another approach is to use use an interpretability approach to handcraft an input for a network that functioned as an adversarial example (Carter et al., 2019; Casper et al., 2023c; Mu & Andreas, 2020; Hernandez et al., 2022).

**Handcrafting a network that reconstructs a network behavior**: If our explanations for network behavior are sufficient, we should be able to use them to build replacement parts for the original network. Cammarata et al. (2020) validated their interpretation of a curve detector's function in a convolutional neural network by substituting its parts with simple handcrafted replacements.

**Testing on ground truth**: If the weights of a toy neural network were handcrafted by humans, it is possible to obtain a ground truth explanation for how it works. This proves useful for testing explanations produced by interpretability methods. For example, Conmy et al. (2023) validated a tool's ability to attribute model behaviors to internal components by running it on a simple model that implemented a known algorithm. (See also Section 2.1.4).

**Using the hypothesis to achieve particular engineering goals**: Another way to test explanations is to assess their utility in downstream applications (Doshi-Velez & Kim, 2017; Casper et al., 2023a). For example, Templeton et al. (2024) discussed examples where manually editing a large language model based on an interpretation led to predictable high-level changes in its behavior. Meanwhile, Marks et al. (2024) showed how an interpretability tool could assist humans with debugging a classifier in a toy task. Farrell et al. (2024) use unlearning (Section 3.2.2) to demonstrate that learned SDL latents don't quite match human concepts, and might not be optimal for particular downstream use cases, highlighting potential issues with SDL.

**Using the hypothesis to achieve specific engineering goals competitively**: Achieving not only useful, but competitive methods sets an even higher standard. For interpretability tools, the highest evaluation criteria require fair comparisons against relevant baselines on real-world tasks instead of cherry-picking them. However, the practice of conducting evaluations using non-cherry-picked tasks remains relatively uncommon. Although attempts have been made to use techniques in the current interpretability toolkit in such evaluations, they have not proven to be consistently useful (Adebayo et al., 2020; Denain & Steinhardt, 2023; Casper et al., 2022b; Hase et al., 2023; Durmus et al., 2024). Some of the most promising research directions are to use interpretability methods to achieve things that would be hard or impossible to achieve without them (Schut et al., 2023). Unless interpretability methods demonstrate that they are competitive with alternative approaches to achieve engineering goals, then the act of demonstrating their usefulness may lead to a bias toward developing methods that only perform well in best-case scenarios and on simple tasks, rather than those that can handle worst-case scenarios and practical challenges.

Interpretability researchers have historically faced challenges in adequately validating their hypotheses due to the high costs in terms of time and cognitive labor. In the following section, we explore two potential solutions that could simplify the validation process: model organisms and interpretability benchmarks.

**'Model organisms' facilitate hypothesis validation.** Although interpretability is sometimes motivated by achieving engineering goals, it is often also approached through the perspective of the natural sciences (Olah et al., 2020b). In certain natural sciences, such as genomics and neuroscience, it is common for researchers to investigate a few extensively studied species known as 'model organisms' or 'model systems'. By conducting in-depth studies on a select group of organisms, like *E. coli*, fruit flies, mice, and macaque monkeys, researchers can leverage the insights and tools gained from those organisms and apply them to other species. For example, imaging specific types of neural activity in mice is more tractable due to existing hypotheses about which proteins should be fluorescently labeled in order to identify specific types of neurons. The use of model organisms allows for cross-checking results with previous work, enabling stronger validation of hypotheses.

Currently, interpretability researchers lack consensus on which networks should serve as model organisms. Essentially, what should the *Drosophila melanogaster* of mechanistic interpretability be? In mechanistic interpretability, an ideal model organism should be open source, easy and cheap to use, representative of a broad range of systems and phenomena, have a replicable training process with open source training data, and have multiple instances with different random seeds, among other criteria (Sharkey et al., 2022a). Thus far, researchers have mostly used model organisms that possess only some of these criteria, such as a transformer than can perform modular addition (Nanda et al., 2023a) or GPT-2 (Radford et al., 2018).

Model organisms not only support cross-validation of hypotheses, but also facilitate the progressive construction of experimental infrastructure by providing a reliable foundation for experiment design. This simplifies the process of rigorous hypothesis testing, thus helping prevent oversimplification and 'interpretability illusions'.

Studying solely model organisms, instead of more directly pursuing engineering goals, risks merely making true statements about neural network structure, rather generating insights that are of immediate practical benefit. For mechanistic interpretability to make the fastest and most substantial progress toward engineering goals, both scientific and engineering wins should be pursued in parallel.

Furthermore, certain choices made while studying model organisms risk steering the field in suboptimal directions. For instance, interpretability research is often motivated by the engineering goal of understanding state-of-the-art models thoroughly enough to make assurances of their safety (Bereska & Gavves, 2024; Tegmark & Omohundro, 2023; Dalrymple et al., 2024). However, limiting its focus by studying small toy models (e.g. Nanda et al. (2023a)) or how larger models accomplish select subtasks (Arditi et al., 2024), risks incentivizing research and methods that fail to generalize to more safety-relevant real-world settings.

**Validating interpretability methods using benchmarks.** Beyond validating individual hypotheses, we may wish to validate entire interpretability methods. Benchmarking is a proven approach to making

incremental improvements in other areas of machine learning, with several approaches to benchmarking interpretability methods being developed in recent years.

One desideratum for interpretability benchmarks is to evaluate interpretations against ground truth explanations (Freiesleben & König, 2023; Zhou et al., 2022). Benchmarks can be established using models with known ground truth explanations. Such models can be created by compiling simple programs into weights of models that exactly implement the known program (Lindner et al., 2023; Weiss et al., 2021; Thurnherr & Scheurer, 2024; Gupta et al., 2024). Alternatively, predetermined explanations can be enforced at training time in conventional models using Interchange Intervention Training (Gupta et al., 2024; Geiger et al., 2022). Other interpretability benchmarks that evaluate specific steps in the interpretability pipelines also exist, such as model decomposition (Huang et al., 2024a; Makelov et al., 2024), generating descriptions of network component functions (Schwettmann et al., 2023), or testing natural language explanations (Huang et al., 2023). Other benchmarks offer tests for multiple facets of the interpretability pipeline in order to test explanation faithfulness, but may require further adaptation to the context of mechanistic explanations (Hedström et al., 2023).

While progress toward the goals of reverse engineering continues, it must be recognized that it is an ambitious goal, especially for very large neural networks. It may be the case that the field fails ultimately to make progress toward complete reverse engineering because of the fundamental difficulty of understanding such large, complex systems. One manifestation of this might be that, even though we might make progress toward 'carving networks at their joints', the components that result are either too complex or numerous to make ready use of them. Similar concerns have motivated some researchers in computational neuroscience to aim to study the learning process of a neural system, rather than the mechanisms that result (Lillicrap & Kording, 2019).

## 2.2 Concept-based interpretability: Identifying components for given roles

### 2.2.1 Concept-based probes

When attempting to localize a human-interpretable concept within the network, an intuitive approach is to 'probe' for it (Köhn, 2015; Gupta et al., 2015; Alain & Bengio, 2017; Ettinger et al., 2016). A concept-based probe is a classifier trained to predict a concept from the hidden representation of another model (Hupkes & Zuidema, 2018). These concepts are typically human concepts of special interest. Probing requires a labeling function that assigns classification labels to input data, indicating the 'value' of the concept on that data (Note: A binary value indicating the presence or absence of a concept is a special case of this approach). Once the labels are assigned, a probe, which is a simple parameterized model, is trained to predict concept labels based on hidden activations. If the probe is a linear model, then we have localized the concept as a vector in latent space.

Probes were first introduced in NLP (Köhn, 2015; Gupta et al., 2015) and have since been extensively explored in the field (Conneau et al., 2018; Tenney et al., 2019; Rogers et al., 2020; Gurnee et al., 2023; Peters et al., 2018; Burns et al., 2023; Marks & Tegmark, 2024). They have also been applied in vision (Alain & Bengio, 2017; Kim et al., 2018b) and deep reinforcement learning (McGrath et al., 2022; Forde et al., 2023). Probing also includes concept activation vectors (Kim et al., 2018a), information-theoretic probing (Voita & Titov, 2020), and structural probing (Hewitt & Manning, 2019).

Although relatively simple to implement, probing has two main challenges (Ravichander et al., 2021; Belinkov, 2022b): (1) The need for carefully chosen data for well-defined concepts, and (2) probes detect correlations instead of causal variables in hidden activations.

### 2.2.2 Probes need carefully chosen data for well-defined concepts

Concept-based probing requires a labeling function that assigns labels for the (human) concept of interest to input data. Obtaining a labeling function is not always trivial and may require substantial human effort to define a data set for a single concept. Moreover, it is only possible to identify concepts that we have defined precisely enough to create high-quality data. This limitation implies that concept-based probing can only identify concepts that we were already looking for, rather than reveal unexpected features in the

network. Another approach, known as Contrast-Consistent Search (CCS), probes not for single concepts but an axis in activation space that corresponds to positive or negative propositions by enforcing probabilistic consistency conditions (Burns et al., 2023). Despite being more unsupervised than standard concept-based probing, even CCS requires the construction of data sets with clear positive and negative cases. A potential path forward for concept-based decomposition is to develop methods that automatically develop data sets for probing and concept localization (Shaham et al., 2025) (Section 2.4).

### 2.2.3 Probes detect correlations, rather than causal variables, in hidden activations

Probes are tools for a correlational analysis, measuring if hidden activations serve as signals for a given concept. Indeed, from an information-theoretic point of view, an arbitrarily powerful probe measures the mutual information between a hidden representation and a concept (Hewitt & Liang, 2019; Pimentel et al., 2020).

However, training a probe to associate a concept with specific hidden activations does not necessarily imply that those activations causally mediate how that concept is used by the network, or even if the network uses the concept at all (Ravichander et al., 2021; Geiger et al., 2024b; Elazar et al., 2021; Belinkov, 2022b). Probes can be successfully trained on hidden activations that lack any causal connection to the output, only localizing correlated hidden activation vectors. For this reason, probing should be used only to generate hypotheses about which network components might be causally linked to a concept. Confirming such hypotheses requires further investigation using causal interventions or other probing methods.

To improve the causal relevance of probe vectors, one approach is to use counterfactual data, which involves intervening on the concept of interest (for instance, observing the resulting output if the dog in an image was changed to a cat) (Elazar et al., 2021; Mueller, 2024; Geiger et al., 2024b). Methods include distributed alignment search (Geiger et al., 2024c; Wu et al., 2023; Huang et al., 2024a), causal probing (Guerner et al., 2024), using attribution methods to measure the effect of concept vectors on network predictions (Kim et al., 2018b), and various concept erasure methods (Ravfogel et al., 2020; 2022; Elazar et al., 2021; Belrose et al., 2023b;b). While these methods can identify causal mediators of concepts in hidden representations, they require more specialized data than probes do.

At times, it might be acceptable for probes to identify merely correlated hidden activations if the correlations generalize to the test distribution. However, probing approaches face a considerable risk of not only discovering correlated features, but also spurious correlations due to the high dimensionality of hidden activations. Validating probes is therefore essential to avoid overfitting. This includes evaluating them on out-of-distribution test data that varies along task-specific dimensions to ensure that general purpose features have been found. One question that remains unanswered is how to use regularization to achieve good probe generalization.

### 2.2.4 Concept-based intrinsic interpretability

Although probes begin with a trained network and search for specific concepts within it, it is also feasible to leverage the concepts in the network training process itself, such as in the case of concept bottleneck models (Koh et al., 2020). This is beneficial as it is more likely that the components to which concepts are assigned are causally relevant. Instead of specific concepts, networks can also be trained to use particular causal structures (Geiger et al., 2022). While it may not be possible to prespecify all relevant concepts or structures, integrating this approach with methods for disentangling concepts could prove useful (Chen et al., 2018). Cloud et al. (2024) incentivize modularity via applying data-dependent, weighted masks to gradients during backpropagation.

### 2.3 Proceduralizing mechanistic interpretability into circuit discovery pipelines: A case study

How should we codify the process of mechanistic interpretability to yield the deepest possible insights? To form a complete pipeline by combining various methods, several methodological choices regarding decomposition, description, and validation, must be made. Circuit discovery has emerged as a prominent pipeline in recent mechanistic interpretability research (Wang et al., 2023; Hanna et al., 2023; Heimersheim & Janiak,

2023). Its objective is to describe how a neural network performs a task of interest while making specific choices for network decomposition, component description, and hypothesis validation. In this section, we look at the typical choices in each step in greater depth and discuss how this popular pipeline could be improved.

The 'circuit discovery' pipeline takes the following steps:

1. **Task Definition**. For a given model we want to study, we select a task that the model can perform, and a dataset on which the network performs that task. This is a concept-based step, since the definition of the task was based on how human researchers define a task distribution.

2. **Decomposition**. During the decomposition step, it is common to think of the neural network as a directed acyclic graph (DAG), where activations are represented by nodes and the "abstract weights" between them represented by edges. Most work thus far has selected architectural components (Section 2.1.2), such as attention heads and MLP layers, to be the nodes. However, more recent work has also used SDL latents for nodes (Marks et al., 2024)

3. **An initial description step: Identify task-relevant vs. -irrelevant subgraphs.** The circuit discovery procedure then identifies task-relevant nodes and edges. Typically, causal interventions are used, drawing samples from some "clean" and "counterfactual" data sets. Circuit discovery methods are generally based on iterative activation patching (Wang et al., 2023; Chan et al., 2022a; Lieberum et al., 2023) or integrated gradients (Marks et al., 2024).

4. **An iterative description-validation loop**. After obtaining a task-relevant subgraph, the next step involves describing the function of each node or edge individually. This step is less formulaic than previous steps. Researchers rely on their intuition, attempting to create testable hypotheses for the function of a component or edge of the circuit, and then design custom experiments to validate or invalidate their hypothesis. Only after several iterations of hypothesis testing through experimentation are researchers finally satisfied with their explanation. In research papers, this loop is rarely made explicit, as only the final description is presented. However, Chan et al. (2022b) detail this process for understanding the induction task, and Nanda (2023b)provides another description of such a loop (building on work by Li et al. (2023)).

5. **Final Validation**. Circuits are commonly evaluated based on three attributes (Wang et al., 2023): *faithfulness*, which refers to how closely the circuit approximates the entire network's behavior, *minimality*, which assesses if nodes in the subgraph are unnecessary, and *completeness*, which determines whether any nodes not included in the subgraph are important for task behavior. Additional ad hoc validation methodologies also exist. For example, Wang et al. (2023) generate adversarial examples for their task based on their mechanistic understanding, while Shi et al. (2024) devise a suite of formal statistical hypothesis tests for circuit efficacy.

This circuit discovery procedure has yielded valuable insight, but falls short using current methods. The pipeline has several issues:

**Task definition is concept-based**. Defining circuits has thus far been with respect to tasks defined by humans. Miller et al. (2024) demonstrate that the within-task variance of model performance across the distribution of data points in a task is large, implying that the circuit provides a good approximation of the average case performance on the dataset, but a poor one for any individual data point. This suggests that the process of first selecting a task and then studying how the model performs it may not be an effective approach to achieve "reverse engineering" -style interpretability. This approach may also create spurious correlations, an issue shared with concept-based probing (Section 2.2.1). Thus, it might be worth learning the task decomposition instead (Haani et al., 2024).

**Network decomposition methods are flawed**. Perhaps most importantly, prior circuit discovery work has attempted to decompose models in either architectural bases (Wang et al., 2023; Conmy et al., 2023) or sparse autoencoder latents (Marks et al., 2024; Huben et al., 2024), which are imperfect ways to decompose

neural networks for mechanistic interpretability (Section 2.1.2). Future work could locate circuits in improved decompositions or simultaneously learn both network decompositions and circuits.

**Circuit faithfulness is low**. Simple early circuits were found to be unfaithful (Chan et al., 2022b; 2023). Miller et al. (2024) show that existing measures of faithfulness depend on the causal intervention implementation used, and further demonstrate that such metrics are misleading when applied to several complex end-to-end circuits. Makelov et al. (2023) argue that subspace activation patching via distributed alignment search may lead to interpretability illusion mechanisms, although these findings are contested by Wu et al. (2024c).

**Scalable methods are only approximate**. Identifying relevant components through individual interventions is costly when there are many components. Attribution patching Syed et al. (2024) was designed to identify potential relevant candidates for further testing through intervention, which becomes more important as the number of components expands significantly through sparse dictionary learning (Marks et al., 2024). However, attribution patching uses gradients, which only yield a first-order approximation of the effect of ablating components (Wu et al., 2024c; Molchanov et al., 2017), leaving it unclear whether this method and any improvements on it (Kramár et al., 2024) produce adequate approximations.

**Circuit discovery algorithms struggle with backup and negative behavior**. Additional challenges for circuit analysis arise from the effects of "backup" and "negative" behavior (Wang et al., 2023; McGrath et al., 2023; McDougall et al., 2024), which actively suppress task performance and are thus not captured by maximizing task performance metrics. Despite this, they remain important factors to consider; Mueller (2024) provides further discussion of these issues.

**'Streetlight' Interpretability**: Interpretability that focuses on models or behaviors that seem easier to interpret than average can be termed 'streetlight interpretability', after the 'streetlight effect'. Much work in mechanistic interetability so far arguably belongs in this category, since the tasks studied so far have been deliberately selected to be simple to define and study mechanistically (Wang et al., 2023). This gives a misleading impression of the level of difficulty involved in implementing circuit discovery for any arbitrary task that a network implements. Indeed, attempts to study arbitrary circuits have proceeded less successfully (Nanda et al., 2023b).

Solving issues with current mechanistic interpretability pipelines remains an open challenge that promises significant benefit. Upon establishing reasonable procedures, automating the overall pipeline will become more feasible. However, some individual steps in mechanistic interpretability can already be fruitfully automated, as discussed in the next section (Section 2.4).

## 2.4 Automating steps in mechanistic interpretability research

Historically, mechanistic interpretability research has required considerable manual researcher effort, though it typically studies models that are smaller than those at the frontier. To make interpretability useful for downstream use cases, scalable approaches are crucial. In this section, we discuss *automated interpretability* methods. We will explore previous cases where manual tasks in mechanistic interpretability have been successfully automated and address open problems in further automation.

**Automating feature description and validation**. A task that is amenable to automation is 'describing the functional role of model components' (Section 2.1.3). With the increasing sophistication of language models, researchers have generated descriptions of the functional role of neurons in image models (Hernandez et al., 2022), neurons in language models (Bills et al., 2023), and sparse autoencoder latents in language models (Huben et al., 2024; Bricken, 2023; Juang et al., 2024) using highly activating data set examples. These interpretations are validated by assessing how effectively a human or model can use them to predict the activation of a feature in a given data set example, or predict where a feature is active within a single image or text excerpt. The success of these predictions can be used as a quantitative measurement of 'interpretability'. This was previously used to measure progress toward a decomposition method that carves networks at the joints of their generalization structure, assuming that such a decomposition would be maximally interpretable (Section 2.1.2). While imperfect, these methods for interpretation hypothesis generation and validation might be improved by automating the generation of inputs to test the interpretation hypotheses by ensuring that

generations activate the interpreted feature (Huang et al., 2023), or defining more rigorous statistical tests (Bloom & Lin, 2024b). Automatic component labeling could expand in the future to include descriptions of feature effects, relationships between features (Bussman et al., 2024), or how components interact during runtime produce behavior.

**Automating circuit discovery pipelines**. An approach called Automated Circuit DisCovery' (ACDC) automates part of the pipeline discussed in Section 2.3 to identify computational subgraphs involved in particular tasks (Conmy et al., 2023). Several works have since improved upon and accelerated this process (Syed et al., 2024; Kramár et al., 2024; Marks et al., 2024). Note that ACDC-like approaches in general assist in identifying relevant subgraphs for a pre-defined task, but do not automate important subsequent steps, such as describing the functional role of subgraph components.

While significant progress has been made toward automating steps of mechanistic interpretability pipelines, fully automating current pipelines would not yield satisfactory explanations of model behavior[6]. Further methodological progress is required for fully automated neural network interpretability to be capable of generating the quality of interpretations necessary to achieve our goals.

---

[6]For one attempt at this using leading decomposition and description methods, see Marks et al. (2024)

# 3 Open problems in applications of mechanistic interpretability

Ultimately, we need mechanistic interpretability methods that enable us to solve concrete scientific and engineering problems. While predicting the impact of fundamental science in advance is difficult, having concrete goals in mind during research is usually beneficial. We want mechanistic interpretability methods to help us achieve various outcomes, such as monitoring and auditing AI systems more effectively (Section 3.2), controlling of AI system behavior more precisely (Section 3.2.2), predicting AI system outcomes more accurately (Section 3.3), enhancing AI system capabilities (Section 3.4), and extracting knowledge from AI systems (Section 3.5). We should also anticipate that mechanistic interpretability will uncover "unknown problems" present in systems, revealing that the true realm of challenges and possibilities is greater than what we currently perceive it to be.

As highlighted in the previous section, progress in mechanistic interpretability methods is multifaceted. Each axis of methodological advancement leads to varying degrees of progress toward different goals. Before we discuss open questions in its applications, we identify distinct axes of methodological progress that lead to different amounts of progress toward different goals.

## 3.1 Axes of mechanistic interpretability progress.

**Decomposition vs. description of network components**: Improvements in network decomposition versus component description methods offer varying benefits for different goals. Decomposition methods vary in their efficacy at carving networks at the joints of their generalization structure, while description methods can yield descriptions that vary in depth. Deeper descriptions of a component are typically more causal or mechanistic, whereas shallower descriptions may rely more on correlations and only connect to inputs or outputs without referencing intermediate causes or effects (Figure 6). Deeper descriptions thus attempt to explain more about how the component interacts with other components within the network's algorithm. Certain goals can be achieved with minimal or no progress in decomposition or description, while others may demand substantial progress.

**Extent of network decomposition or description**: The extent of network decomposition or description needed may vary depending on the goal. Certain goals only require an understanding of specific network components (such as an individual features or a circuit), while others might require enumerating or understanding of larger circuits or the entire model (as in 'enumerative safety').

**Extent of task distribution analyzed**: The scope of task distribution analysis also depends on the intended goal. For instance, monitoring a model for a single kind of behavior might only require decomposing or understanding the model only over a narrow task distribution, while others, such as formal verification, might demand understanding over the entire distribution of tasks.

**Mechanistic understanding post vs. during training**: Understanding the mechanisms of a fixed model might suffice for some goals, but more ambitious goals might require an understanding of not only the models' mechanisms, but also how they change during the learning process.

In this section, we'll discuss how mechanistic interpretability has been used or could be leveraged to further the field's various goals. We'll assess the progress made thus far, and identify the advancements along different axes of methodological progress that will be most crucial to success.

## 3.2 Using mechanistic interpretability for better monitoring and auditing of AI systems for potentially unsafe cognition

### 3.2.1 Mechanistic interpretability-based evaluations could help us detect unsafe or unethical AI cognition

Currently, we rely on "black box" evaluations to understand a model's capabilities, but studying input-output behavior alone may not reveal all dangerous behaviors. Such behaviors include deceiving users (Park et al., 2023b; Ward et al., 2023; Scheurer et al., 2024; Meinke et al., 2024); for instance, by intentionally underperforming on evaluations ("sandbagging", van der Weij et al. (2024)), leveraging situational awareness

(Laine et al., 2024); or giving dishonest responses tailored to match the user's beliefs ("sycophancy"; Sharma et al. (2024b)). Interpretability techniques could be used to uncover the mechanisms underlying these potentially harmful behaviors and thus help to detect and characterize them. This becomes increasingly important as the capabilities of models increases, especially when using training methods that incentivize models to feign particular properties for the purpose of passing evaluation.

Using interpretability methods to identify internal signs of concern (also known as "white-box" evaluations (Casper et al., 2024) or "understanding-based" evaluations (Hubinger, 2023)) is therefore an important problem. White-box evaluation methods could serve as tools to detect potential biases that arise when models learn to use spurious correlations (Gandelsman et al., 2024a; Casper et al., 2022a; Abid et al., 2022). However, human judgment might be required to determine which features are 'supposed' to be relevant to the task (Marks et al., 2024; Kim et al., 2018a; Goyal et al., 2022).

Despite current shortcomings in decomposition and description, white-box evaluations are likely feasible today. Even shallow, correlation-based descriptions could signal potentially concerning cognition. For example, developing new methods that reliably distinguish between features that merely recognize deceptive behavior vs. mechanisms that cause deceptive behavior may be challenging. However, a correlation-based method that flags both can facilitate catching the latter. To be useful, it may not even be necessary to decompose or describe the entire network; having descriptions for components that are used on concerning subdistributions of model behavior might suffice. For instance, imperfect interpretability methods may help evaluators develop hypotheses about how models will behave, thus guiding further inquiry. Meanwhile, recent work proposes incorporating SDL (Section 2.1.2) into safety cases for advanced AI systems. By monitoring internal representations, it could aid in detecting potential sabotage or deceptive behavior before deployment (Grosse, 2024). While such approaches show promise, they have difficulty in validating whether learned features capture all concerning patterns of reasoning reliably.

Evaluations for unsafe cognition may be a particularly important use case as it plays well to the comparative advantages of mechanistic interpretability relative to the other areas of machine learning. The majority of other areas of machine learning already focus on controlling or steering the behavior of AI systems to alter input-output behavior. It is therefore unclear that this is to mechanistic interpretability's comparative advantage. On the other hand, interpretability is perhaps the only research area that attempts to understand the mechanisms of model cognition. This implies that it might be particularly fruitful for interpretability researchers to tackle problems that become easier to solve through improving such understanding: auditing for unsafe cognition, debugging unexpected behavior, and monitoring systems in deployment.

**Enabling real time monitoring of AI systems for potentially unsafe cognition**   Beyond white-box evaluations, interpretability has further applications in monitoring. For instance, internals could be used to passively monitor the system during deployment, much like content moderation systems currently in use today. Alternatively, internals could be used to flag when a model takes an action for abnormal reasons even in absence of satisfactory descriptions (Section 2.1.3), known as "mechanistic anomaly detection" (Christiano, 2022; Johnston et al., 2024), which may be a sign of suspicious behavior. Mechanistic anomaly detection primarily requires progress in decomposition methods (Section 2.1.2) as it is necessary to be confident about what constitutes an individual mechanism within the network. Current SDL methods identify active latents, but not active mechanisms, which are implemented by network parameters. It may not be necessary to have deep descriptions of the function of individual mechanisms as long as anomalies can be detected.

**Improving our ability to red-team AI systems and elicit unsafe outputs**   Beyond white-box evaluations, leveraging interpretability could improve our ability to conduct adversarial attacks, red-team, or jailbreak AI systems (Casper et al., 2024). This process is beneficial as it exhibits failure modes models may display in the wild, when facing adversarial pressure from wide deployment or malicious actors, thereby enabling developers to effectively preempt and address them. Furthermore, it may form a significant element of safety cases (Clymer et al., 2024; Balesni et al., 2024a; Grosse, 2024; Goemans et al., 2024) for AI systems, by providing assurances of form: We tried hard to red-team the system, yet failed to exhibit concerning behavior despite having *more* affordances than users may have. One reasonable assumption is that developers may have white-box access to models, while users may not. Although many existing red-teaming

methods (Perez et al., 2022; Zou et al., 2023b) already require gradient access, we could additionally leverage interpretability insights to accelerate red-teaming. For instance, Arditi et al. (2024) discovered a universal "refusal direction" in chat-finetuned language models, which is causally important for models engaging in the behavior of refusing harmful requests. Lin et al. (2024) used this to red-team models by optimizing for inputs that minimize the projection of the residual stream onto this direction during the forward pass. This approach may be more efficient than optimizing over the whole model, as previous methods did (Zou et al., 2023b). Mechanistic interpretability techniques also promise to improve our ability to attribute model outputs to their corresponding inputs (Section 2.1.3). This could enhance human red-teamers' ability to find key input features responsible for bad behavior in models, thus speeding up iteration cycles. Though such approaches are possible today with only feature-based understanding, they might improve with more crisp mechanism-based understanding.

### 3.2.2 Using mechanistic interpretability for better control of AI system behavior

Ensuring the safe deployment of AI first requires effective control over their behavior. Currently, the techniques used for this purpose are mostly unrelated to interpretability (e.g. Christiano et al. (2017); Rafailov et al. (2023); inter alia), but sometimes inspired by it (Rimsky et al. (2024), Zou et al. (2024), Kirch et al. (2024); inter alia). Mechanistic interpretability could assist in interpreting (Lee et al., 2025) and improving (Conmy & Nanda, 2024) these control methods, or in developing new ones. In this section, we outline interpretability-inspired control methods, and envision future possibilities with further progress in interpretability methods.

One new control method derived from mechanistic interpretability insights is **activation steering** (a.k.a. activation addition). A fixed activation vector, hypothesized to linearly represent a model concept, is added to an intermediate activation of a model at inference time (Li et al., 2024a; Turner et al., 2024; Zou et al., 2023a; Rimsky et al., 2024). Turner et al. (2024) introduced activation steering, directly inspired by the Linear Representation Hypothesis (discussed in Section 2.1.2). This technique results from the hypothesis, and its success can be thought of as evidence for the hypothesis. Moderate success can be achieved in steering using basic decomposition and description methods. Mechanistic interpretability decomposition methods enable the steering of models toward a narrower range of behaviors with fewer side effects (Chalnev et al., 2024). Advancements in mechanistic interpretability methods are likely to result in improved steering capabilities, such as activating entire mechanisms instead of individual features.

**Machine unlearning** was originally defined as the problem of scrubbing the influence of particular data points on a trained machine learning model (Cao & Yang, 2015). In the context of modern generative models, machine unlearning is more broadly defined as removing particular undesirable knowledge or capabilities ('unlearning targets') from models, while preserving model performance on tasks involving non-targets (Liu et al., 2024a). Targets for unlearning that are of particular interest include sensitive private or copyrighted data, model biases (Liu et al., 2024b), and hazardous knowledge that could be misused by malicious actors (Li et al., 2024b); for instance, information regarding the creation of bioweapons. A better understanding of how knowledge or capabilities are implemented within model internals can help in the development of new techniques for machine unlearning (Belrose et al., 2023b; Zou et al., 2024; Guo et al., 2024; Pochinkov & Schoots, 2024; Ashuach et al., 2024), as well as to better evaluate unlearning efficacy through white-box, non-behavioral, techniques (Lynch et al., 2024; Deeb & Roger, 2024; Hong et al., 2024). Thus far, mechanistic interpretability methods that modify intermediate activations (but not weights) for unlearning have yet to yield competitive results (Farrell et al., 2024).

Unlearning falls under the broader aim of **model (knowledge) editing**, which seeks to make precise modifications to a machine learning model that incorporates specific knowledge with desirable generalization properties, while minimizing the impact on other knowledge (Wang et al., 2024c). By attempting to carve neural networks at their joints, mechanistic interpretability could improve our ability to make interventions on knowledge with few side effects. Meng et al. (2022a) make initial progress toward interpretability-based model editing with their ROME technique. However, Thibodeau (2022) and Hase et al. (2023) highlight flaws in the technique, indicating that mechanistic interpretability has not yet found appropriate model components to intervene on (Section 2.1.2). With better comprehension of neural networks, we should anticipate more surgical model editing techniques in the future.

Editing any given capability or piece of knowledge presents a greater challenge than deleting them. Meaningful progress in unlearning and editing methods may depend on improved network decomposition methods, as it would require isolating the individual mechanisms that correspond to specific knowledge or capabilities. Progress in mechanistic interpretability may elucidate the structure of knowledge and capabilities in AI models, leading to a better understanding of what kinds of model edits possibilities are realistic in future. Knowledge and capabilities could, in fact, be part of large mechanisms that overlap with each other, making it challenging to isolate them into discrete components. For mechanistic interpretability to effectively guide editing, strong description methods will be necessary to understand how to modify specific targets without affecting others.

Finally, mechanistic interpretability may provide tools to rigorously **understand how finetuning alters models**. This may assist in debugging instances in which finetuning leads to undesired and spurious effects (Casper et al., 2023b). Recent work (Jain et al., 2024; Prakash et al., 2024; Lee et al., 2025) suggests that existing finetuning methodologies primarily make shallow edits to existing model representations and circuitry. Importantly, this suggests that harmlessness training (which trains models to refuse to answer harmful requests) may be cheaply undone. Empirical evidence supports this claim, both with further finetuning (Gade et al., 2024; Lermen et al., 2024), as well as with causal interventions on the forward pass (Arditi et al., 2024). Separately, localizing knowledge and capabilities within models may improve the sample efficiency of finetuning, by selectively modifying only relevant parameters (as in, e.g. Wu et al. (2024b)). Further advancement in tools for comparing feature-level differences between models (such as Lindsey et al. (2024)) may accelerate our ability to debug finetuning or other control methods (Bricken et al., 2024). Mechanistic interpretability work has thus yielded several insights into how finetuning changes models and how to improve it, and may provide further insights and improvements in the future.

### 3.3 Using mechanistic interpretability for better predictions about AI systems

Accurately predicting model behavior in new scenarios or regimes is difficult (arguably impossible) without understanding model internals. Interpretability could hopefully facilitate two kinds of predictions:

- Predicting model behavior in novel situations

- Predicting capabilities that arise during training or finetuning

#### 3.3.1 Predicting behavior in novel situations

In order to determine whether an AI system may potentially underperform poorly or pose a safety risk in new situations, the ability to predict its behavior in untested settings is imperative. A model's behavior, which may only become apparent in unforeseen circumstances, cannot be fully captured by its performance on a finite set of behavioral evaluations.

By understanding the mechanisms of jailbreaking, we can anticipate the means through which a user might bypass existing safeguards (Lee et al., 2025; Arditi et al., 2024). Similarly, if models have "trojans", backdoors (Hubinger et al., 2024), adversarial examples, or biases, comprehending a model's internal mechanisms could improve our ability to predict when models will display undesirable behavior, even if these scenarios were not encountered during standard training or behavioral evaluations. Casper et al. (2023a) benchmark feature synthesis tools through their ability to aid developers in identifying trojans, while interpretability assisted in identifying cases of adversarial examples (Gandelsman et al., 2024b; Mu & Andreas, 2020; Wang et al., 2023; Kissane et al., 2024a), and SDL was used to uncover biases based on spurious correlations in an LLM-based classifier (Marks et al., 2024). Beyond specific failures, interpretability methods can also be used to gain a broader understanding of network behavior. For example, prior work identified signatures in model internals that predict a model's likelihood of hallucinating (Yu et al., 2024) or its knowledge of particular facts (Gottesman & Geva, 2024). Mechanistic interpretability might also be able to identify when networks have learned 'shortcuts' (Geirhos et al., 2020) that might lead to generalization failures under distributional shifts, which may result in unexpected and undesirable behavior. Identifying these 'shortcuts' in advance may help predict these failures in advance (Lapuschkin et al., 2019).

Generally, being able to predict an AI system's behavior in advance is more challenging – but also more desirable – than merely being able to monitor its behavior and cognition. Mechanistic interpretability could allow us to make a certain type of claim, namely, "there exists no mechanisms that would cause the model to deliberately behave undesirably" (Olah, 2023). For a strong version of this claim, substantial progress in both decomposition and description methods is necessary. However, weaker versions of the claim, addressing specific undesirable behaviors, might be more feasible with near-term methods. For instance, if it is possible to decompose networks and identify all components, even basic description methods might let us recognize that there are no mechanisms relating to bioweapons or illicit substances within the network, thus letting us predict that models are probably not capable of instructing users how to fabricate bioweapons or illicit substances (a possibility sometimes referred to as "enumerative safety" (Elhage et al., 2021; Olah, 2023)). As AI systems become increasingly agentic, claims about even more general behavior may be possible. Understanding their values or goals (or, less anthropomorphically, 'the internal mechanisms that determine their action plans and actions') should enable us to better predict their behavior across a broad range of contexts (Colognese & Jose, 2023).

When deploying AI in high-stakes scenarios, rigorous and reliable predictions are necessary, much like those demanded of safety-critical software applications. Sometimes, such software is formally verified, thereby ensuring certain safety-critical aspects of its behavior are guaranteed, since its compliance with specific properties is mathematically proven. In the context of mechanistic interpretability, the equivalent would be formal verification of AI systems (Dalrymple et al., 2024; Tegmark & Omohundro, 2023; Critch & Krueger, 2020): mathematically proving that an AI system's behavior will satisfy a desired property on any input in a given distribution. Formal verification of AI systems remains an unresolved issue at present. The level of understanding of AI necessary to enable formal verification of large, general AI systems for nontrivial properties is well beyond the current capabilities of mechanistic interpretability. However, some recent studies using toy models provide a glimpse into what solving formal verification of AI might look like. Approaches inspired by mechanistic interpretability have been used to prove accuracy bounds on a single-layer transformer trained on a synthetic task, albeit with great difficulty (Gross et al., 2024). Program synthesis through mechanistic analysis offers an alternative approach by converting simple trained neural networks into more interpretable, controllable, and verifiable programs (Michaud et al., 2024).

Several open questions remain about the tractability of scaling these approaches from toy models to frontier systems. For instance, for program synthesis, it is uncertain to what extent computations within real-world neural networks can be reduced to operations that can be cleanly represented in symbolic code, or what the total length of such code would be. Ensuring the safety of agentic systems with formal guarantees is further complicated by the need to model a system's interactions in an arbitrarily complex environment that might not be formalizable (Seshia et al., 2022; Wongpiromsarn et al., 2023; Dalrymple et al., 2024).

### 3.3.2 Predicting capabilities that arise during training or finetuning

The most competitive methods of AI development systems result in uninterpretable systems that often fail in ways that surprise their developers (OpenAI et al., 2024; Team et al., 2024; Anthropic, 2024). Applying mechanistic interpretability to alleviate this issue is a key area for future research.

Improved mechanistic understanding of model training could enhance the ability to predict when certain capabilities will appear. For instance, it has been observed that new model capabilities can emerge as a function of scale (Wei et al. (2022), though also see Schaeffer et al. (2023)). Evidence suggests that new capabilities may be learned in a somewhat discrete Michaud et al. (2023) or stagewise (Wang et al., 2024b) fashion, and that in synthetic data settings, the emergence of new capabilities coincides with abrupt changes in the trajectory of model parameters (Park et al., 2024a).

Other work shows a correlation between in-context learning capabilities and the emergence of induction heads, an attention-based circuit mechanism (Olsson et al., 2022). By connecting these threads of research, the long-term hope for mechanistic interpretability research is to link small-scale mechanistic structure to larger-scale structure, such as the evolving shape of the loss landscape during model scaling (Olah, 2023). To make progress toward this goal, research needs to move beyond simply improving 'decomposition' (Section 2.1.2)

or 'description' (Section 2.1.3) quality and instead be capable of describing the dynamic changes in the mechanistic structure of networks throughout the learning process.

We may also want to link the emergence of capabilities to specific properties of the training data set. Through mechanistic interpretability, we can create data sets that facilitate training models to demonstrate desirable attributes and predict their behavior. By attributing model outputs to specific training examples, influence functions have been applied to LLMs (Koh & Liang, 2017; Grosse et al., 2023) to predict limitations in their generalization abilities, such as a lack of robustness when the order of certain phrases was flipped (Berglund et al., 2024). Other work examines how data set composition shapes the emergence of in-context and weights-based learning (Reddy, 2024).

A related problem of interest involves predicting which model capabilities, that may not be present in a given model, can be "elicited" with sufficiently advanced prompting or finetuning strategies (Greenblatt et al., 2024). Prakash et al. (2024) find evidence that finetuning improves capabilities primarily by enhancing existing circuits, rather than developing fundamentally new mechanisms. Relatedly, (Jain et al., 2024) and Lee et al. (2025) show that finetuning can mask capabilities present in a base model in a way that can easily be reversed via simple changes to the model. Improved mechanistic understanding of finetuning could help reveal capabilities obscured in this fashion. Since capabilities are behaviors that often span multiple sequential steps, it may be necessary to have mechanistic interpretability methods that examine mechanisms spanning multiple time steps. However, current mechanistic interpretability research is primarily focused on understanding mechanisms involved in predictions at a single time step.

### 3.4 Using mechanistic interpretability to improve our ability to perform inference, improve training, and make better use of learned mechanisms

A mechanistic understanding of AI models could be leveraged to improve their utility, from faster inference and better training, to enhancing and manipulating representations.

By understanding the internal generation process of AI models, we could accelerate their inference. For example, it could help identify which parts of the computation could be skipped without changing the model's final output (Voita et al., 2019; Din et al., 2024; Voita et al., 2024; Gromov et al., 2024). The ability to inspect the information or functions implemented in a model could facilitate the development of more effective distillation methods by recognizing gaps that should be distilled (Gottesman & Geva, 2024) and discovering novel ways to distill them (Zhang et al., 2024).

Another aspect that mechanistic interpretability could enhance is model training. Interpreting how the model processes specific examples and using this information to influence its predictions (Koh & Liang, 2017; Grosse et al., 2023) may inform the selection of better training data to improve the model's capabilities in desired ways. Moreover, better monitoring of the training process can be achieved by correlating certain drops in training loss with capability gains (Olsson et al., 2022; Wang et al., 2024a) or identifying a general order in which specific capabilities emerge during training (Michaud et al., 2023). In addition, identifying the contributing components of a given task could help to devise novel, parameter-efficient training methods. Finally, being able to decompose networks into their functional components presents possibilities to build components that lend themselves to learning computational structures that we better understand (Crowson et al., 2022; Fu et al., 2023).

Mechanistic interpretability has the potential to not only accelerate AI inference and training, but also enhance its utility. Intervening in the model's computation has the potential to remove unwanted bugs in its reasoning abilities, and achieve better balance between its knowledge recall process and latent reasoning (Yu et al., 2023; Jin et al., 2024; Biran et al., 2024; Balesni et al., 2024b). More broadly, understanding the inner workings of different models could lead to better recombination of what they have learned, such as combining model parameters (Wortsman et al., 2022) and transferring representations across models (Ghandeharioun et al., 2025; Csiszárik et al., 2021).

### 3.5 Using mechanistic interpretability for 'microscope AI'

Current approaches for knowledge discovery from data involve statistical or causal analysis, dimensionality reduction, or using machine learning models that are inherently interpretable. These techniques can be valuable, but are influenced by human priors, typically assume linear relationships between variables, and cannot handle massive multimodal data. On the other hand, deep learning models are capable of encoding complex, non-linear relationships and extracting meaningful features from massive data sets without human intervention. Historically, these abilities had limited scientific value, as without methods to interpret these models, we could not understand the patterns they found. However, with ongoing advancements in interpretability research, this is beginning to change.

Applying interpretability for knowledge discovery is sometimes called **microscope AI**. This approach involves training a neural network to model a data set, then applying interpretability techniques to the model to gain insight into any (potentially novel) predictors it discovers. In this way, the superhuman pattern matching skills of deep neural networks can serve as a tool to parse complex data sets. By understanding these networks' detailed mechanisms, mechanistic interpretability may add new details to our understanding of the domains in which they have learned to perform so well.

Current methods allow for versions of microscope AI, depending on the kind of insights that we want to learn. Some examples of these applications include extracting novel chess concepts from AlphaZero and teaching them to top grandmasters (Schut et al., 2023) using a CNN trained on defendant mugshots and judge decisions to reveal how facial features affect judgments (Ludwig & Mullainathan, 2023), transforming psychology articles into a causal graph with an LLM to enable link prediction and produce expert-level hypotheses (Tong et al., 2024b), analyzing a CNN to learn previously unknown morphological features for predicting immune cell protein expression (Cooper et al., 2022), and applying methods like BiLRP to assess similarity and extract insights from historical documents such as astronomical tables (Eberle et al., 2022), among several other studies (O'Brien et al., 2023b; Hicks et al., 2021; Narayanaswamy et al., 2020; Korot et al., 2021). As interpretability methods improve along various axes, deeper insights in and across more domains will become possible.

Currently, the majority of scientists are unable to access microscope AI due to the need for specialized expertise in machine learning, interpretability, and domain knowledge to recognize significant new patterns, a combination of skills that is rare in many fields. This may change as interpretability research becomes more widely adopted in the sciences and as interpretability becomes increasingly automated and accessible.

### 3.6 Mechanistic interpretability on a broader range of models and model families

The vast majority of mechanistic interpretability research to date has focused on just three model families: CNN-based image models (e.g. Erhan et al. (2009); Nguyen et al. (2016c); Olah et al. (2020b)), BERT-based text models (e.g. Devlin (2018); Rogers et al. (2020)) and GPT-based text models (e.g. Elhage et al. (2021); Wang et al. (2023); Nanda et al. (2023a)). The degree of generalizability of these findings to other models and contexts is currently a somewhat open question. Given that future frontier models may use architectures that differ from the current state-of-the-art, and are expected to be multimodal by default, interpretability researchers may need to expand the range of models and modalities that they study and try to identify universal approaches that can be applied to all of them.

Assessing how well interpretability methods apply to architectures beyond those for which they were developed, and whether we can develop techniques that generalize effectively across architectures remain open questions. This is especially important due to the recent success of other competitive architectures as alternatives to CNNs and transformers. Notable alternatives include diffusion models (Sohl-Dickstein et al., 2015; Rombach et al., 2022) and Vision Transformers (Dosovitskiy et al., 2021) for image generation/classification and RWKV (Peng et al., 2023) and later state space models (SSMs) (Gu & Dao, 2024) for language modeling. Recent studies show that certain methods transfer from CNNs to SSMs (Paulo et al., 2024), and from transformers to some SSMs e.g. (Meng et al. (2022b) vs. Sharma et al. (2024a)) and (Wang et al. (2023) vs. Ensign & Garriga-Alonso (2024)).

Beyond the transferability of interpretability methods, a related open question concerns the transferability of conclusions across model families. The overwhelming majority of mechanistic interpretability research focuses on the transformer model family and therefore does not distinguish between observations that are model-specific and those that are not. Consequently, we may be overlooking valuable insights that could be obtained by comparing the results across multiple model families. These insights may, for example, evince or refute the 'universality hypothesis', which states that (Li et al., 2015; Olah et al., 2020b) different neural networks learn similar features and circuits to one another.

### 3.7 Human computer interaction with model internals

As we saw in Section 3.3, the ability to control and understand neural networks is tightly linked. Thus, mechanistic interpretability has great potential to facilitate new types of human-AI interaction. Systems amenable to human comprehension and control would allow diverse users to intuitively manipulate and interact with them based on their preferences, greatly broadening their utility. As a starting point, AI engineers who build and test AI systems have an obvious interest in working with the internal workings of neural nets. If experts could visualize and interact with internal representations, it would unlock obvious benefits for scientific research.

However, interactive tooling has a much broader constituency, including policy-focused AI auditors, as well as end users. Consider an auditor looking for bias or safety issues in a neural net. With a way to probe the network directly instead of relying on testing behavior, the auditor can be much more successful in discovering potential low-probability but high-stakes errors. For end users, transparency into a network facilitates appropriate calibration of trust. One recent idea proposes a dashboard that can show, in real time, the internal features that influence a chatbot's answers during a text chat (Chen et al., 2024; Zou et al., 2023a; Viégas & Wattenberg, 2023). Such a dashboard might help users spot AI errors, or display warnings if safety-relevant features activate. More generally, results from mechanistic interpretability could be used to blend direct-manipulation interfaces with text interfaces, providing users with a richer palette of controls (Carter & Nielsen, 2017).

# 4 Open socio-technical problems in mechanistic interpretability

Effective practical application of mechanistic interpretability brings both technical challenges and complex social ramifications. It could enable us to act on AI policy and governance, presenting a valuable opportunity to implement regulatory standards and social ideals through technical means (Section 4.1). Such consequential impacts inevitably come with important social and philosophical considerations, which likewise require rigorous inquiry if we are to fully realize the potential benefits of AI (Section 4.2).

## 4.1 Translating technical progress in mechanistic interpretability into levers for AI policy and governance

Current frontier AI governance efforts rarely specify concrete ways in which a mechanistic understanding of AI models might be used to help implementation. For example, OpenAI's Preparedness Framework commits to mitigating biological risks posed by its AI systems (OpenAI, 2023), but the framework lacks details on specific measures that might be taken. Progress in interpretability could potentially enable the removal of any knowledge from the model which could aid users in creating biological weapons (Li et al., 2024b). However, it remains uncertain how technical progress will translate into better AI governance, largely due to the numerous open technical problems in mechanistic interpretability. However, there are several promising routes toward better levers for AI policy and governance.

These avenues include assisting companies and governments to identify risks through evaluations and enhancing forecasts about new AI developments; more thorough oversight of AI systems in deployment; simplifying how AI systems operate within existing liability law through clearer explanations of AI decisions; enabling governments to establish risk mitigation regulations and companies to commit to concrete mitigation commitments; and protecting copyright law.

An understanding of model internals can help AI labs assess the risks from frontier models (Chang et al., 2024; Shevlane et al., 2023; Casper et al., 2024) and thus better fulfill their obligations under the EU AI Act to "perform model evaluation ... with a view to identifying and mitigating systemic risk" (Council of the European Union, 2024). More specifically, a mechanistic understanding of models could help evaluators elicit dangerous capabilities via improved finetuning (United Kingdom AI Safety Institute, 2024), guide their adversarial red-team attempts (Tong et al., 2024a), and ensure that AI systems are not intentionally underperforming evaluations (van der Weij et al., 2024). Advancements in mechanistic interpretability could also assist companies and governments in anticipating when or if AI models will obtain specific dangerous capabilities. Improved forecasting capabilities could enhance threat modeling by reducing the "reasonable disagreement amongst experts over which risks to prioritize" (Anthropic, 2024). For instance, it may help build consensus on whether language models are just stochastic parrots (Bender et al., 2021) or if they have coherent world models (Li et al., 2023). We also might be able to use interpretability to build evidence supporting or refuting different models of catastrophic threat, such as determining the validity of mesa-optimization (Hubinger et al., 2021) or inner alignment concerns (Carlsmith, 2023). It would also give additional time for companies to prepare adequate risk mitigation measures, (OpenAI, 2023), and for governments to establish appropriate guidance or regulation (UK Government, 2022).

The EU AI Act mandates that developers of General Purpose AI models with systemic risk have obligations to report incidents involving their system to the AI office. Interpretability tools have the potential to continuously monitor AI inference and detect incidents that require reporting. Compared to incidents in other domains (for instance, nuclear security), AI systems allow us to log all inputs and system states, even those that may lead to catastrophic harm. Access to a small number of such data points may greatly improve our ability to mitigate similar future failures (Greenblatt & Shlegeris, 2024). For example, interpretability could be used to investigate the critical "features" of the input that led to the AI incident. This could improve our ability to further red-team the system (Section 3.3) and generate more similar data points that could result in similar incidents. This, in turn, may help us reduce the likelihood of future incidents (Chan et al., 2024). The incident could be utilized more directly in the construction of test-time monitors to detect similar future incidents (see Roger (2023)). More speculatively, interpretability could be used to verify companies' compliance with domestic regulation (O'Brien et al., 2023a), or to authenticate states' compliance with future international treaties regarding the use of AI (Aarne et al., 2024).

Leveraging a mechanistic understanding of model internals could also make decision rationales for AI model outputs more easily obtainable. This could aid in enforcing citizens' rights under the EU General Data Protection Regulation "to obtain an explanation of the decision reached" by a system "based solely on automated processing" (GDP, 2016; Gilpin et al., 2019). Model editing tools could also resolve problems regarding the copyright status of existing generative models (Grynbaum & Mac, 2023). According to the US Copyright Act (United States Congress), for any copyrighted work, an artifact "from which the work can be perceived, reproduced, or otherwise communicated... with the aid of a machine or device" is considered a copy of the work (Lee et al., 2024). Interpretability tools could help detect and remove memorized works that can be reproduced verbatim by generative models.

## 4.2 Social and philosophical problems in mechanistic interpretability

The ability to interpret advanced AI systems holds immense potential to advance the science of AI and increase our ability to control it (Critch & Krueger, 2020; Tegmark & Omohundro, 2023; Dalrymple et al., 2024). In this paper, we provide an overview of various interpretability tools that offer novel insights. Nonetheless, interpretability research has thus far produced few tools that are used to make state-of-the-art systems safer in the real world (Rauker et al., 2023). Modern AI systems are still generally trained, evaluated, monitored, and debugged using techniques that do not rely on understanding their internal workings.

The absence of paradigmatic clarity is a major socio-technical factor for this. Questions such as which goals the field of interpretability should pursue, how success should be graded, and how we should define interpretability warrant more thoughtful answers than exist at present. In interpretable AI research, the motivations and methods employed are often described as "diverse and occasionally discordant" (Lipton, 2018). At times, the objective of AI interpretability research is articulated as advancing a fundamental "understanding" or "uncovering the true essence" (Christensen & Cheney, 2015) of what is happening inside black-box models. The intrinsic validity of this paradigm deserves philosophical inquiry.

However, from an engineer's perspective, pursuing "understanding" without a practical downstream application misses an engineer's objective. Proponents of this view may contend that quantifiable benchmarks linked to concrete practical goals are accurate measures of the success of interpretability. This motivation is concrete and useful, but some have criticized interpretability research as artificially limiting the solution space to engineering problems. When interpretability tools are studied with motivations such as fairness or safety, Krishnan (2020) argues that, "Since addressing these problems need not involve something that looks like an 'interpretation' (etc.) of an algorithm, the focus on interpretability artificially constrains the solution space by characterizing one possible solution as the problem itself." Thus, some argue that interpretability research has failed to produce competitive techniques (Rauker et al., 2023; Casper et al., 2022b), omitted non-interpretability baselines (Rudin, 2019; Krishnan, 2020), and graded interpretability tools on their own curve (Doshi-Velez & Kim, 2017; Miller, 2019; Rauker et al., 2023). In all safety-relevant applications of mechanistic interpretability, it is important to assess the usefulness of interpretability against alternative methodologies. Failing to do so or misrepresenting these comparisons can lead to follow-up work that rests on false assumptions (Lipton & Steinhardt, 2019; Leech et al., 2024).This is particularly problematic when it impacts the efforts of critical safety work.

Despite these concerns, it is not always imperative for mechanistic interpretability to strictly outperform uninterpretable baselines: It may still be helpful to develop methods that offer interpretability-based advantages, along with the benefits of more competitive uninterpretable methods (e.g. Rimsky et al. (2024)). While interpretability-based methods do not currently outperform black-box baselines, if they perform in a similar ballpark, further progress in mechanistic interpretability could soon lead to better methods, especially in problems which we believe might disproportionately benefit from improved mechanistic understanding (Section 3).

Another potential reason for lack of clarity is that models are often studied in a vacuum: Smart & Kasirzadeh (2024) emphasize that the usefulness or correctness of model interpretations can depend on the broader context of the model's development or deployment. For example, it may be hard to identify representations of fairness within models, since understanding this requires an understanding of broader contexts. The same data may lead to different conclusions under different definitions of fairness.

Finally, the interpretability community must exercise caution in their communication to minimize potential abuses of the results of its work. Unfortunately, selective transparency can be used to actively mislead (Ananny & Crawford, 2018). Furthermore, interpretability is at risk of being used for purposes that might serve corporate interests at the potential expense of safety. The field of AI interpretability is highly influenced by research teams in the private sector. On one hand, industry resources and research contributions have significantly advanced interpretability research. On the other hand, compared to academia, corporations publish selectively, often have financial conflicts of interest, and may provide limited transparency. Meanwhile, the recent definite — but ultimately modest — progress in mechanistic interpretability has been used to lobby against specific AI regulation by falsely claiming that, "Although advocates for AI safety guidelines often allude to the 'black box' nature of AI models, where the logic behind their conclusions is not transparent, recent advancements in the AI sector have resolved this issue, thereby ensuring the integrity of open-source code models." (Andreessen Horowitz, 2023).

## 5 Conclusion

While mechanistic interpretability has made meaningful progress in both methods and applications, significant challenges remain before we can achieve many of the field's ambitious goals, if they can be achieved.

The path forward requires progress along multiple axes. We would benefit from stronger theoretical foundations for decomposing neural networks at the joints of their generalization structure. Current methods like sparse dictionary learning, while promising, face both practical limitations in scaling to larger models and deeper conceptual challenges regarding their underlying assumptions. We must also develop more robust methods for validating our interpretations of model behavior, moving beyond correlation-based descriptions to capture true causal mechanisms. Additionally, we need better techniques to understand how mechanisms evolve during training and how they interact to produce complex behaviors.

These methodological advances could unlock several promising applications. Improved interpretability methods could enable more effective monitoring of potential risks, better control over model behavior, and more accurate predictions of capabilities. For AI capabilities, mechanistic understanding could lead to more efficient architectures, better training procedures, and more targeted ways to enhance model performance. In various scientific domains, microscope AI approaches could help extract valuable insights from model internals. To achieve these diverse goals, the field must ensure a focus on generating insights that have real-world utility. This will involve establishing better benchmarks and comparing interpretability-based approaches to non-interpretability baselines. However, fastest progress will likely come from the field pursuing both scientific and engineering goals simultaneously, rather than one at the expense of the other.

The practical impact of progress in mechanistic interpretability extends beyond technical achievements. Interpretability tools could provide crucial mechanisms for governance and oversight. They could help verify compliance with safety standards, detect potential risks before deployment, and provide clearer attribution of model decisions. However, realizing these benefits will require careful attention to the risks of potential misuse and of giving false assurance about AI safety.

Looking toward the future, many expect current AI capabilities to be only a foretaste of what is to come. As AI capabilities advance, the need for a mechanistic understanding of their decision-making processes becomes increasingly urgent. While the black-box nature of AI models remains unresolved, the untapped potential of mechanistic interpretability is what makes it such an exciting research area, and highlights the importance of solving its many open research problems.

**LS** managed the project and made major contributions to planning, writing, and editing content, as well as coordinating other contributors and synthesizing their perspectives where necessary. **BC** substantially contributed to the framing, writing and editing of the final manuscript, and made all of the figures. All authors (**LS**, **BC**, **JBn**, **JL**, **JW**, **LB**, **NGD**, **SH**, **AO**, **JBm**, **SB**, **AGA**, **AC**, **NN**, **JR**, **MW**, **NS**, **JM**, **WS**, **EJM**, **SC**, **MT**, **DB**, **ET**, **AG**, **MG**, **JH**, **DM**, **TM**) contributed to the initial writing and editing of various sections and giving feedback on versions of the manuscript.

We also greatly thank Fayth Tan for assistance with editing and Schmidt Sciences for their feedback at various stages of the project and for funding the work.

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

# Appendix

# Summary of open questions

**Open problems in mechanistic interpretability methods and foundations**

**Reverse engineering: Identifying the roles of network components**

**Reverse engineering step 1: Neural network decomposition**

1. How should we decompose networks into more interpretable constituent parts?

   (a) What isomorphism or what approximation of a neural network (or parts of it) is the best way to express it for the purposes of interpreting it?
   (b) How should we coarse grain neural networks?
   (c) How should we build higher level abstractions on top of low-level network components?

2. How true is the linear representation hypothesis?

   (a) To what extent do models encode concepts linearly in their representations?
   (b) How should we characterize representations that are not linearly represented in neural networks?
   (c) What properties of a concept, or of the training distribution, result in a particular concept becoming encoded linearly (or not)?

3. Is the combination of the linear representation hypothesis and superposition the right frame for thinking about computation in neural networks?

   (a) Can we fully determine the causes of feature superposition and polysemanticity within neural networks?
   (b) How should we understand superposition in attention blocks?
   (c) How should we understand cross-layer superposition?
   (d) What new theoretical insights can be gleaned from considering how networks perform computation natively in superposition, rather than treating superposition purely as a compression strategy?

4. Can the problems with SDL be overcome?

   (a) What lies in SDL reconstruction errors? Will the errors converge to zero with methodological progress?
   (b) Is sparsity the correct proxy for interpretability?
   (c) Can the approach be scaled to the largest models?
   (d) Does SDL make sense if we don't believe in the linear representation hypothesis?
   (e) Is sparsity the best possible proxy for interpretability?
   (f) Is it correct to think of SDL features as 'bags of features', or is there important information contained within the geometry of representation space?
   (g) If SDL finds compositions of the "true" features, is this a problem?
   (h) Can SDL be applied to all architectural components to fully decompose networks?
   (i) How can we better measure the success of SDL techniques?
   (j) How should we connect sparsely activating features into circuits? Will this be practically possible, and the best possible description of network mechanisms?
   (k) Can we develop new methods that address conceptual and practical issues with SDL?

5. How important is the geometry of activation space for explaining neural network behavior?

   (a) How can we identify the underlying functional structure of networks (which defines why activations are located in particular geometric arrangements in activation space)?

    (b) Must we understand global feature geometry or only local feature geometry in order to understand computation in neural networks?

6. Can we connect theories for how neural networks generalize to interpretability?

    (a) Can we distinguish parts of networks that underlie generalization from parts that underlie memorization?

    (b) What mechanisms underlie the relationship between interpretability and generalization?

    (c) Are there connections between adversarial robustness and superposition?

    (d) Can we connect interpretability to theories of deep learning like SLT?

7. Can we build intrinsically more interpretable models at low performance cost? How helpful is this?

    (a) Interpretability training: Can we train networks that are interpretable by default at low performance cost?

    (b) Interpretable inference: Can we convert already-trained models into forms that are much easier to completely interpret at little performance cost?

    (c) How can we train large-scale models such that the concepts they use are naturally understandable to humans?

    (d) How can we design training objectives so that the model is incentivized to use known specific abstractions?

    (e) How can we localize concepts we want to control (e.g., long-term plans) during training?

**Reverse engineering step 2: Identifying the functional role of components**

1. Can we improve on max-activating input data set examples for understanding the causes of network component activations?

    (a) How can we avoid imposing human bias to explanations?

    (b) Can we progress toward deeper descriptions based on internal mechanisms?

    (c) How might we develop interpretation methods that can recognize and work with unfamiliar concepts - computational patterns that don't map cleanly to human intuitions?

2. How can we develop attribution methods that faithfully and efficiently compute which network components are important for some downstream metric?

    (a) How can we develop attribution methods that capture higher-order effects beyond first-order approximations of model behavior?

    (b) Is it possible to create perturbation-based methods that don't force models to operate outside their training distribution?

    (c) Can we develop hybrid approaches that combine the strengths of different attribution methods while mitigating their individual weaknesses?

3. How can we better measure the downstream effects of model components?

    (a) How can we reliably distinguish between true causal pathways and compensatory effects like the "Hydra effect" when performing interventions?

**Reverse engineering step 3: Validation of descriptions**

1. Can we improve our ability to validate mechanistic explanations for model behavior in ways that do not depend on researcher intuition and are computationally tractable to use?

    (a) Can we improve on methodologies for evaluating hypotheses through their predictive power on activations of network components?

    (b) Can we develop methodologies for evaluating hypotheses through their predictive power for model behavior (e.g. unusual failures or adv examples)?

    (c) Can we develop methodologies for handcoding weights that faithfully represent some hypotheses, as a drop in replacement for subnetworks we claim to understand?

    (d) Can we develop a wider suite of networks with known ground truth explanations to validate techniques against?

    (e) Can we use mechanistic explanations to achieve engineering goals?

    (f) Can we use mechanistic explanations to achieve engineering goals in a way that improves upon black box baselines?

2. Can we develop "model organisms" as a community, which are understood deeply, and seen as a test-bed for new unproven interpretability methodologies to be tested?

3. Can we establish standardized baselines and benchmarks for comparing different interpretability approaches on real-world, non-cherry-picked tasks, where the ground truth is known?

4. What would constitute a comprehensive set of "stress tests" for interpretability hypotheses that could reliably detect interpretability illusions?

5. How might we design evaluation frameworks that assess interpretability methods on their average case and worst-case performance rather than just best-case scenarios?

6. How can we ensure that our understanding of internals generalizes to out-of-distribution inputs?

**Concept-based interpretability: Identifying network components for given roles**

1. How can we reliably distinguish causal from merely correlated features when probing neural networks?

2. Can we develop automated systems to generate high-quality probing data sets, reducing the current heavy reliance on human effort?

3. What regularization and validation techniques can be used to prevent spurious correlations while ensuring probes find generalizable features?

4. How can we improve probing for concepts that may not have clear positive/negative examples?

**Proceduralizing mechanistic interpretability into circuit discovery pipelines**

1. Can we develop techniques that build on lower level methods that provide deeper or more complete insights about neural networks?

2. How much can we learn from further work in the existing circuit discovery paradigm?

    (a) Should we expect circuit discovery to benefit from further methodological progress in decomposing neural networks? Will faithfulness go up and explanation description length go down?

    (b) Can we remove the constraint that task definition for circuit discovery is inherently concept-based, which may be complicating mechanistic analysis?

    (c) Can we get around the practical issues of negative and backup behavior?

    (d) Will circuit discovery provide insights into arbitrary tasks, or will it only be helpful in cases where we are able to crisply define tasks?

**Automating steps in mechanistic interpretability research**

1. Can we improve on AI automated feature description and validation methods?

    (a) Through automating the generation and testing of arbitrary hypotheses?

    (b) Through describing differences between features?

    (c) Through descriptions of how components interact?

2. Can we improve on ACDC-like circuit discovery methods?

3. Can we automate other parts of the mechanistic interpretability pipeline?

    (a) Conceptual interpretability research?
    (b) Decomposition method discovery?
    (c) More ad hoc validation of hypotheses?

4. Should we take steps to mitigate potentially misaligned AI systems sabotaging AI automated interpretability?

**Open problems in applications of mechanistic interpretability**

**Using mechanistic interpretability for better monitoring and auditing of AI systems for potentially unsafe cognition**

1. Can we effectively use interpretability for safety evaluations?

    (a) Can we develop robust "white box" evaluations that detect concerning internal patterns without needing to understand the entire network?
    (b) Can we reliably distinguish between features that merely recognize deceptive behavior versus mechanisms that generate deceptive behavior?
    (c) How can we validate that learned features capture all concerning patterns of reasoning?
    (d) Can we reliably identify which features are appropriately versus spuriously relevant to a given task?

2. Can we leverage interpretability to enhance red-teaming and system testing?

    (a) Can we use interpretability insights to make red-teaming more efficient than current methods?
    (b) How can we best use feature attribution to help human red-teamers identify problematic input patterns?

3. Can we develop effective test-time monitoring systems based on interpretability?

    (a) Can we get mechanistic anomaly detection to work?
    (b) Can we create passive monitoring systems based on model internals that effectively flag concerning internal patterns during deployment?
    (c) Can we develop monitoring systems that work with only feature-level understanding rather than requiring deep mechanical insights?

**Using mechanistic interpretability for better control of AI system behavior**

1. Can we improve steering methods through interpretability?

    (a) How can we make activation steering more precise and reduce its side effects?
    (b) Can we develop methods to steer entire mechanisms rather than just single features?

2. Can we achieve reliable model unlearning and editing?

    (a) Will carving the network at its true joints help us improve on model unlearning and editing?
    (b) Can mechanistic interpretability help us develop better methods for evaluating unlearning efficacy?
    (c) Can mechanistic interpretability help us determine which classes of model edit are even possible, without damaging generalization in undesirable ways?

3. Can we better understand and improve finetuning through interpretability?

    (a) Can we make finetuning more sample-efficient by targeting specific parameters?
    (b) Can we develop better tools for analyzing feature-level or mechanism-level differences between model versions?

**Using mechanistic interpretability for better predictions about AI systems**

1. Can we predict model behavior in novel situations outside of the distribution of inputs we have access to with mechanistic understanding?

   (a) Can we reliably predict when and how jailbreaking or safety bypasses might occur?

   (b) How can we identify internal signatures that predict specific failure modes like hallucination?

   (c) Can we develop methods to predict model behavior without requiring behavioral evaluations?

   (d) Can we find "values" or "goals" in systems that might be indicative of behavior in more generality?

   (e) Is it possible to prove the absence of specific dangerous capabilities through mechanistic analysis?

2. Can we develop formal verification methods for AI systems?

   (a) Can current toy model verification approaches scale to frontier systems?

   (b) How much of neural computation can be reduced to verifiable symbolic operations?

   (c) Can we create formal guarantees about system behavior in complex, non-formalizable environments?

   (d) What level of mechanistic understanding is necessary for meaningful formal verification?

3. Can we make rigorous claims about model safety?

   (a) Can we definitively prove the absence of specific dangerous mechanisms?

   (b) How can we verify claims about model values and goals in a rigorous way?

   (c) What types of safety claims are possible with current interpretability methods?

   (d) Can we develop "enumerative safety" approaches that reliably identify all relevant mechanisms?

4. Can we better predict AI capability development through interpretability?

   (a) Can we identify early signatures that predict emergent capabilities?

   (b) How do model mechanisms evolve dynamically during training?

   (c) Can we map the connection between small-scale circuits and large-scale capabilities?

   (d) How does the loss landscape's structure relate to capability emergence?

5. Can we understand the relationship between training data and capabilities?

   (a) How do specific training examples influence the development of model mechanisms?

   (b) Can we predict model limitations based on training data composition?

   (c) Can we design training data sets to reliably produce specific desired capabilities?

   (d) How does data set structure affect the balance between in-context and weights-based learning?

6. Can we predict latent or maskable capabilities?

   (a) How can we identify capabilities that could be 'unlocked' through prompting or finetuning?

   (b) Can we detect when finetuning has masked rather than removed capabilities?

   (c) How do we analyze mechanisms that span multiple timesteps or sequential behaviors?

   (d) Can we predict which model capabilities are fundamental versus superficially trained?

**Using mechanistic interpretability to improve our ability to perform inference, improve training and make use of learned representations**

1. Can we use interpretability to make inference more efficient?

   (a) How can we identify skippable computations without affecting outputs?

   (b) Can we create more effective distillation methods through mechanistic understanding?

   (c) How can we optimize model architecture based on component function analysis?

(d) Can we identify and optimize critical computational pathways?

2. Can we improve training through mechanistic insights?

   (a) Can we better select training data by understanding example influence?
   (b) How can we monitor and optimize capability emergence during training?
   (c) Can we develop more parameter-efficient training methods through component analysis?
   (d) Can we create better architectures through component understanding?
   (e) Can we identify and enhance components with specific functionalities?

3. Can we instill capabilities directly into networks?

   (a) Can we design better inductive biases based on mechanistic insights?
   (b) Is it possible to create modular architectures with swappable components?
   (c) Can we develop reliable methods for combining model parameters?
   (d) Is it possible to transfer specific capabilities between models?

**Using mechanistic interpretability for 'microscope AI'**

1. Can we leverage AI models for scientific discovery?

   (a) How can we extract novel patterns and predictors that models have found?
   (b) Can we make microscope AI techniques accessible to domain experts?
   (c) How do we validate scientific insights derived from model interpretability?
   (d) Can we extend microscope AI beyond current simple correlational discoveries?

2. Can we develop better knowledge extraction methods?

   (a) How can we detect when models have found genuinely novel patterns?
   (b) Can we automate the process of finding scientific insights in model weights?
   (c) How do we bridge the gap between model features and scientific concepts?
   (d) Can we make these techniques usable without deep machine learning expertise?

**Mechanistic interpretability on a broader range of models and model families**

1. Can interpretability methods generalize across architectures?

   (a) Do current interpretability methods (SDL, circuit analysis) transfer to SSMs? Or, like the transition from CNNs to transformers, are new approaches necessary?
   (b) Which insights are model-specific versus universal?
   (c) How can we adapt methods for multimodal models?

2. How do different models trained on similar data compare mechanistically?

   (a) Is the "universality hypothesis" true across models? To what extent do neural networks learn similar features and circuits to each other (and to humans?)
   (b) Do different architectures learn fundamentally different features?
   (c) How do mechanisms of particular tasks differ between transformers, CNNs, and SSMs?
   (d) Are there insights we can gain from comparing architectures?

3. Can we future-proof interpretability research?

   (a) How can we prepare for interpreting novel architectures?
   (b) Should we focus on architecture-specific or general methods?
   (c) Can we identify truly fundamental interpretability principles?
   (d) Will current methods work on future frontier models?

**Human computer interaction with model internals**

1. Can we create interfaces that use mechanistic understanding to enhance human-neural network interaction?

   (a) How can we visualize model internals in an intuitive way?
   (b) Can we develop real-time interpretability dashboards?
   (c) What's the right balance between simplicity and depth in these interfaces?
   (d) How do we make complex model mechanisms understandable to non-experts?

2. Can we develop interpretability tools to help auditors?

   (a) How can we help auditors find potential failure modes directly?
   (b) Can we develop tools to detect bias at the mechanism level?
   (c) What interfaces would make auditing more efficient and thorough?
   (d) How can we present technical findings to policy makers?

3. Can we improve end-user interaction with AI?

   (a) How can transparency features help users calibrate trust?
   (b) Can we create intuitive controls based on model mechanisms?
   (c) Can we create intuitive ways to steer model behavior?

**Governance**

1. Can mechanistic analysis help identify and prevent failures?

   (a) Can we identify specific mechanisms that caused AI failures?
   (b) How do we map the causal chain of mechanisms leading to incidents?
   (c) Can we detect when similar mechanisms are about to activate?
   (d) Is it possible to isolate and modify failure-causing mechanisms?

2. Can we study mechanism patterns related to governance?

   (a) Can we identify mechanisms responsible for specific dangerous capabilities?
   (b) How do we detect deceptive or evasive mechanisms?
   (c) Can we map the mechanisms involved in model decision-making?
   (d) Is it possible to verify the absence of specific harmful mechanisms?

3. Can mechanistic insights verify compliance?

   (a) How can we trace decision mechanisms to explain model outputs?
   (b) Can we identify mechanisms that process copyrighted content?
   (c) Is it possible to detect mechanisms that encode specific knowledge?
   (d) How do we verify modifications to problematic mechanisms?

**Open socio-technical problems in mechanistic interpretability**

**Translating technical progress in mechanistic interpretability into levers for AI policy and governance**

1. Can we use a mechanistic understanding to better evaluate AI capabilities?

   (a) How can we use interpretability to improve capability elicitation?
   (b) Can we use interpretability to reliably detect when models are strategically underperforming capabilities evaluations?

2. Can we use a mechanistic understanding to improve our ability to forecast when or whether new capabilities will arise ahead of time?

3. How can we use interpretability to better estimate the likelihood of different threat models?

4. Can we use interpretability to prevent AI incidents?

   (a) Can we use interpretability to construct reliable test-time monitors to detect AI incidents?
   (b) Can we use reliably prevent similar incidents in the future, by using interpretability to design new evaluation tasks on incident scenarios?

5. Can interpretability help verify which workloads GPUs are being used for?

6. How should interpretability inform copyright law?

7. How can mechanistic understanding help resolve copyright challenges in generative AI, particularly regarding the detection and removal of memorized copyrighted works?

**Social and philosophical challenges in mechanistic interpretability**

1. What is interpretability?

   (a) What are the goals of the field?
   (b) How should success be graded?
   (c) Should we treat interpretability as a science or an engineering discipline? What implications does this have on what research should be done?

2. How can we mitigate downside risks of interpretability research?

   (a) How can we communicate the results of our research such that the risk of their misuse is minimized?

Table 1: Summary of principal methodologies in mechanistic interpretability (Section 2)

| Methodology Category | Principal Method / Approach | Description / Goal | Key Techniques / Examples | Major Open Problems / Limitations (Sec. 2) |
|---|---|---|---|---|
| **Reverse Engineering** | **1. Neural Network Decomposition** | Breaking down the network into simpler, analyzable parts to understand how it generalizes (Section 2.1.2). | <ul><li>Architectural components (neurons, layers) (Section 2.1.2)</li><li>Dimensionality reduction (PCA, SVD) (Section 2.1.2)</li><li>Sparse Dictionary Learning (SDL) (SAEs, Transcoders) (Section 2.1.2)</li><li>Intrinsically decomposable models (e.g., sparse activations, modular training) (Section 2.1.2)</li></ul> | <ul><li>Architectural components often polysemantic (Section 2.1.2).</li><li>Dim. reduction can't find more features than dimensions (Section 2.1.2).</li><li>SDL: High reconstruction error, expensive, assumes linear representation, sparsity as proxy for interpretability is questionable, feature geometry unexplained, doesn't decompose mechanisms themselves (Section 2.1.2), lack of solid theoretical foundations (Section 2.1.2).</li><li>Intrinsic methods: Performance costs, superposition can "sneak through" (Section 2.1.2).</li></ul> |

Table 1: Summary of principal methodologies in mechanistic interpretability (Section 2) (Continued)

| Methodology Category | Principal Method / Approach | Description / Goal | Key Techniques / Examples | Major Open Problems / Limitations (Sec. 2) |
|---|---|---|---|---|
| | **2. Describing Component Functional Roles** | Formulating and hypothesizing the functional role of decomposed components and their interactions (Section 2.1.3). | • *Causes of activation*: Highly activating examples, attribution methods (Grad-CAM, Integrated Gradients, SHAP, LIME), feature synthesis (Section 2.1.3). 
 • *Downstream effects*: Logit lens, causal interventions (patching, ablation), observing sequential behavior (steering, patchscopes) (Section 2.1.3). | • Highly activating examples: Human bias, interpretability illusions, correlational (Section 2.1.3). 
 • Attribution: Often first-order approximations, can be misleading or manipulated, perturbation issues (Section 2.1.3). 
 • Feature synthesis: May not reflect natural data, struggle with trojans (Section 2.1.3). 
 • Logit lens: Measures direct effect only (Section 2.1.3). 
 • Causal interventions: Can be expensive (Section 2.1.3). 
 • Chain-of-thought: Not always faithful to internal process (Section 2.1.3). |

Table 1: Summary of principal methodologies in mechanistic interpretability (Section 2) (Continued)

| Methodology Category | Principal Method / Approach | Description / Goal | Key Techniques / Examples | Major Open Problems / Limitations (Sec. 2) |
|---|---|---|---|---|
| | **3. Validation of Descriptions** | Testing if hypotheses about component functions are correct and distinguishing faithful from merely plausible explanations (Section 2.1.4). | • Predicting activations/counterfactuals (Section 2.1.4).
• Explaining unusual failures/adversarial examples (Section 2.1.4).
• Handcrafting network replacements (Section 2.1.4).
• Testing on ground truth models (Section 2.1.4).
• Achieving engineering goals (competitively) (Section 2.1.4).
• Using 'model organisms' (Section 2.1.4).
• Interpretability benchmarks (Section 2.1.4). | • Difficulty distinguishing faithful from plausible explanations (Section 2.1.4).
• Interpretability illusions (Section 2.1.4).
• High cost (time, cognitive labor) for validation (Section 2.1.4).
• Lack of consensus on model organisms (Section 2.1.4).
• Current benchmarks may need adaptation for mechanistic explanations (Section 2.1.4).
• Risk that complete reverse engineering is too complex/numerous components (see end of Section 2.1.4). |

| Methodology Category | Principal Method / Approach | Description / Goal | Key Techniques / Examples | Major Open Problems / Limitations (Sec. 2) |
|---|---|---|---|---|
| **Concept-Based Interpretability** | **Concept-based Probes** | Identifying network components that correspond to pre-defined, human-interpretable concepts (Section 2.2.1). | <ul><li>Linear probes (Section 2.2.1).</li><li>Concept Activation Vectors (Section 2.2.1).</li><li>Information-theoretic probing (Section 2.2.1).</li><li>Structural probing (Section 2.2.1).</li><li>Contrast-Consistent Search (CCS) (Section 2.2.2).</li><li>Causal probing variants using counterfactual data, distributed alignment search (Section 2.2.3).</li></ul> | <ul><li>Requires carefully chosen data for well-defined concepts (human effort) (Section 2.2.2).</li><li>Detects correlations, not necessarily causal variables (Section 2.2.3).</li><li>Risk of spurious correlations due to high dimensionality (Section 2.2.3).</li><li>Causal variants require more specialized data (Section 2.2.3).</li></ul> |
| | **Concept-based Intrinsic Interpretability** | Building models that are inherently interpretable with respect to specific human concepts from the training process itself (Section 2.2.4). | <ul><li>Concept Bottleneck Models (Section 2.2.4).</li><li>Training for particular causal structures (Section 2.2.4).</li><li>Gradient routing (Section 2.2.4).</li></ul> | <ul><li>May not be possible to prespecify all relevant concepts/structures (Section 2.2.4).</li></ul> |

Table 1: Summary of principal methodologies in mechanistic interpretability (Section 2) (Continued)

| Methodology Category | Principal Method / Approach | Description / Goal | Key Techniques / Examples | Major Open Problems / Limitations (Sec. 2) |
|---|---|---|---|---|
| **Proceduralizing & Automating Interpretability** | **Circuit Discovery Pipelines** | A structured procedure to identify and describe how a neural network performs a specific task of interest (Section 2.3). | • Steps: Task Definition, Decomposition (DAG), Identify task-relevant subgraphs (e.g., iterative activation patching, integrated gradients), Iterative description-validation, Final Validation (faithfulness, minimality, completeness) (Section 2.3).
• Automated variants like ACDC (Section 2.4). | • Task definition is concept-based, may not be true "reverse engineering" (Section 2.3).
• Relies on flawed network decomposition methods (Section 2.3).
• Low circuit faithfulness in practice (Section 2.3).
• Scalable methods for identifying components (e.g., attribution patching) are approximate (Section 2.3).
• Struggles with "backup" and "negative" behavior (Section 2.3).
• "Streetlight interpretability": focus on simpler, easier-to-interpret tasks/models (Section 2.3). |

Continued on next page

Table 1: Summary of principal methodologies in mechanistic interpretability (Section 2) (Continued)

| Methodology Category | Principal Method / Approach | Description / Goal | Key Techniques / Examples | Major Open Problems / Limitations (Sec. 2) |
|---|---|---|---|---|
| | **Automating Interpretability Steps** | Using AI/LLMs to automate parts of the mechanistic interpretability research process to handle scale and reduce manual effort (Section 2.4). | • Automating feature description and validation (e.g., LLMs explaining neurons/latents based on activating examples) (Section 2.4). 
 • Automating parts of circuit discovery (e.g., ACDC identifying subgraphs) (Section 2.4). | • Current automated methods are imperfect and may require further methodological progress to yield satisfactory explanations (see end of Section 2.4). 
 • Automated circuit discovery doesn't automate describing functional roles of components (Section 2.4). |