# OpenReview forum: "Open Problems in Mechanistic Interpretability"
_TMLR — Accepted by TMLR_

### Review · Reviewer_uHkd · 2025-04-16

**Summary Of Contributions:**

The paper outlines key open challenges in mechanistic interpretability, the study of how neural networks implement computations internally. It highlights two main methodological approaches (reverse engineering and concept-based interpretability) and discusses their current limitations, such as the shortcomings of sparse dictionary learning. The authors emphasize the need for better theoretical foundations, scalable tools, and validation techniques to reliably understand and control model behavior, especially in the context of safety-critical applications.

**Audience:**

No

**Broader Impact Concerns:**

I have no serious concerns regarding broader impact.

**Claims And Evidence:**

No

**Requested Changes:**

I think in its current state, the paper is of limited interest to the ML/TMLR community as it remains fairly superficial (see Weanesses above) with regards to methodological machine learning research and instead focuses on providing a summary of mech interp in 2025.
I have summarized below the key changes that, in my opinion, must be addressed carefully. Collectively, these issues would require a **major revision** of the current draft:

- Provide concrete methodological proposals for advancing mechanistic interpretability.

- Strengthen the connection between the scientific motivation and the rest of the paper, Explainable AI for the Sciences and what does mech interp really contribute to deepening our understanding of AI from first principles?

- Expand the methodological summary beyond the reverse engineering vs. concept-based distinction.

- Clarify the role of human vs. non-human concepts in concept-based interpretability.

-  Include a more extensive discussion on evaluation frameworks and metrics like faithfulness and plausibility, especially in reference to more than a decade of research in developing faithful attribution approaches.

- Remove/revise Figure 6 to convey more meaningful visual information.

- Discuss alternative methods to sparse dictionary learning and relate to prior work in sparse coding and source separation, should this direction be pursued?

- Broaden the discussion on undesired black-box behavior.

- Prioritize methodological grounding as a central challenge for the field.


Some additional related works that were missed

Section 2.1.2
  - Gradient-based Explanations
    - [Bae10] Baehrens, D., Schroeter, T., Harmeling, S., Kawanabe, M., Hansen, K., & Müller, K. R. (2010). How to explain individual classification decisions. The Journal of Machine Learning Research, 11, 1803-1831.

  - Theoretical view on decomposition of neural networks
    - Montavon, G., Lapuschkin, S., Binder, A., Samek, W., & Müller, K. R. (2017). Explaining nonlinear classification decisions with deep taylor decomposition. Pattern recognition, 65, 211-222.

Section 2.1.2
  - Observing the effects of components from a graph-perspective, decomposition into Higher-order structure in Graphs and Transformers
    - [Tsa18] M. Tsang, D. Cheng and Y. Liu, "Detecting statistical interactions from neural network weights", Proc. Int. Conf. Learn. Representations, 2018.
    - [Ebe20] Eberle, O., Büttner, J., Kräutli, F., Müller, K. R., Valleriani, M., & Montavon, G. (2020). “Building and interpreting deep similarity models”, IEEE Transactions on Pattern Analysis and Machine Intelligence, 44(3), 1149-1161.
    - [Jan20] J. D. Janizek, P. Sturmfels and S. Lee. "Explaining explanations: Axiomatic feature interactions for deep networks", CoRR, 2020.
    - [Sch21] Schnake, T., Eberle, O., Lederer, J., Nakajima, S., Schütt, K. T., Müller, K. R., & Montavon, G. (2021). Higher-order explanations of graph neural networks via relevant walks. IEEE transactions on pattern analysis and machine intelligence, 44(11), 7581-7596.
    - [Vas24] A Vasileiou and O Eberle. “Explaining Text Similarity in Transformer Models”, NAACL 2024.
    - [Fum24] Fumagalli, F., Muschalik, M., Kolpaczki, P., Hüllermeier, E., & Hammer, B. (2024). KernelSHAP-IQ: Weighted Least-Square Optimization for Shapley Interactions. arXiv preprint arXiv:2405.10852.

  - Section 2.2.3. The notion of “spurious correlations” seems relevant here.

- Section 2.3. References to works on theoretically grounded subgraph detection in the context of interactions of features/components for interpretability

    - [Sch21] Schnake, T., Eberle, O., Lederer, J., Nakajima, S., Schütt, K. T., Müller, K. R., & Montavon, G. (2021). Higher-order explanations of graph neural networks via relevant walks. IEEE transactions on pattern analysis and machine intelligence, 44(11), 7581-7596.

    - [Fum25] Fumagalli, F., Muschalik, M., Frazzetto, P., Strotherm, J., Hermes, L., Sperduti, A., ... & Hammer, B. (2025). Exact Computation of Any-Order Shapley Interactions for Graph Neural Networks. ICLR 2025.

**Strengths And Weaknesses:**

**Strengths**
  - The paper is overall well-written, easy to follow and discusses the status-quo of mechanistic interpretability.
  - The discussion on sparse dictionary learning (SDL) is detailed and a useful contribution
  - The survey addresses relevant broader scientific, engineering, and societal goals, summarizing the importance of interpretability for the use of AI.

**Weaknesses**
  - The survey mostly focuses on a review of existing problems and approaches that have not worked, but remains quite vague with regard to what methodological approaches could provide a way out. This has been widely recognized as a key issue of mech interp, and the paper does not sufficiently address this challenge.
  - While I appreciate the AI for the Sciences perspective in the introduction, the remainder of the survey does not follow-up on this motivating direction, which makes it appear unconnected. What concrete scientific problems do the authors think can be addressed effectively with mechanistic interpretability (as compared to other interpretability approache and the “Explainable AI for the Sciences” community)
  - The paper lacks a clear methodological summary of the status-quo beyond the vague distinction into  ‘reverse engineering’ and ‘concept-based interpretability’ (Section 2). The focus on the human perspective for ‘concept-based interpretability’ is not clear. Should an interpretability approach not focus on providing any concept, even if not aligned with human perspectives?
  - The survey lacks a dedicated discussion and connection to frameworks developed for evaluating explanations in the context of attributions, e.g. how do commonly used evaluation metrics like “faithfulness” and “plausibility” transfer to mech interp? Section 2.3 touches upon these points (see “5. Final Validation”), but only refers to the limited recent works, see, e.g [Hed23, Vil21] for broader reviews.

    - [Hed23] Hedström, A., Weber, L., Krakowczyk, D., Bareeva, D., Motzkus, F., Samek, W., ... & Höhne, M. M. C. (2023). Quantus: An explainable ai toolkit for responsible evaluation of neural network explanations and beyond. Journal of Machine Learning Research, 24(34), 1-11.
    - [Vil21] Vilone, G., & Longo, L. (2021). Notions of explainability and evaluation approaches for explainable artificial intelligence. Information Fusion, 76, 89-106.

  - Figure 6 does not convey any relevant visual information as it summarizes the headings of Section 3.
  - What’s the use of discussing and dissecting a fairly recent direction of “sparse dictionary learning” and SAEs that has been advocated for by the mech interpretability community from 2021 onwards, to realize that it comes with many flaws that remain difficult to resolve in the context of deep learning models (see Figure 4). One conclusion from this survey may be that this research direction has not been able to fulfil the promises, what are alternatives that may be able to better reconcile these methodological issues. Sparse coding and compressed sensing have a long and theoretically well-grounded history which should be discussed in this context as many of the encountered issues have been identified previously in other contexts, e.g. source separation.
3.2.1
  - Short-sighted survey with respect to undesired black-box behaviors, see e.g. [Lap19, Gei20]

    - [Lap19] Lapuschkin, S., Wäldchen, S., Binder, A., Montavon, G., Samek, W., & Müller, K. R. (2019). Unmasking Clever Hans predictors and assessing what machines really learn. Nature communications, 10(1), 1096.

    - [Gei20] Geirhos, R., Jacobsen, J. H., Michaelis, C., Zemel, R., Brendel, W., Bethge, M., & Wichmann, F. A. (2020). Shortcut learning in deep neural networks. Nature Machine Intelligence, 2(11), 665-673.

  -  The paper lacks a clear prioritization, in my view the main priority should be to ground mech interp in better methodology, yet on this point the paper remains vague and focuses on a summary of all things that have not worked.

---

> ### Author Response · Authors · 2025-05-21
> **Changes made in response to reviewer uHkd feedback (1/3)**
>
> We thank reviewer uHkd for their thorough engagement with our work and for their comprehensive review. We are glad to hear that they found it well written and easy to follow.
>
> We will respond to each of the mentioned weaknesses below, noting any associated Requested Changes.
>
>
> * **Weakness**:
>   > The survey mostly focuses on a review of existing problems and approaches that have not worked, but remains quite vague with regard to what methodological approaches could provide a way out. This has been widely recognized as a key issue of mech interp, and the paper does not sufficiently address this challenge.
>   * **Associated requested change**:
>     > Provide concrete methodological proposals for advancing mechanistic interpretability.
>
>   * **Response**:
>
>
>     We agree that the review does not, for the most part, offer solutions to the many open problems that it lists. We nonetheless think there is substantial value in cataloguing the open problems of the field. While it would be preferable to have solutions, we have not provided them here for a number of reasons. Foremost is that such a wide range of problems would require a wide range of solutions, and for each proposed solution the review would need to make a defensible case. We view this to be outside of the scope of the review, although we would be excited for other reviews with narrower scopes to opine on solutions to individual problem areas that we identify. All this said, we do provide some solutions where those solutions take the form of other open problems (e.g. pointing toward ‘the need for stronger theoretical foundations’ or pointing toward ‘the need for more extensive benchmarking and validation’).
>
>
> * **Weakness**:
>   > While I appreciate the AI for the Sciences perspective in the introduction, the remainder of the survey does not follow-up on this motivating direction, which makes it appear unconnected. What concrete scientific problems do the authors think can be addressed effectively with mechanistic interpretability (as compared to other interpretability approache and the “Explainable AI for the Sciences” community)
>   * **Associated requested change**:
>     > Strengthen the connection between the scientific motivation and the rest of the paper, Explainable AI for the Sciences and what does mech interp really contribute to deepening our understanding of AI from first principles?
>
>   * **Response**:
>
>
>     We think that the reviewer might have missed the section associated with this motivating line in the introduction. The introduction alludes to using MI for knowledge discovery (“What can we learn about protein folding from AIs that can successfully predict protein structure? What insights can we glean about disease from a radiographer that performs beyond human ability?”). We think this is appropriately covered in the section “Using mechanistic interpretability for `microscope AI'”, which details the general uses for MI toward this goal and gives some specific examples. We appreciate that the boundary between ‘mechanistic’ and ‘non-mechanistic’ interpretability is somewhat vague, but if we define ‘mechanistic interpretability’ as interpretability that aims to understand mechanisms on a fine-grained level, as opposed to higher level understanding, it is unclear that non-mechanistic approaches would have the granularity to be able to identify the low-level variables used by the network, which, if humans came to understand them, would comprise ‘knowledge discovery’. This point was previously left implicit in the text, but we have added text to make this more explicit. The text now reads: “... In this way, the superhuman pattern matching skills of deep neural networks can serve as a tool to parse complex data sets. By understanding these networks' detailed mechanisms, mechanistic interpretability may add new details to our understanding of the domains in which they have learned to perform so well.” Beyond this focus on knowledge discovery, which we have now improved, the introduction makes no further implications about AI for the Sciences beyond content that is immediately related to the open problems discussed in the paper, which we would consider outside of the scope of the review.
>
> * **Weakness**:
>   > The paper lacks a clear methodological summary of the status-quo beyond the vague distinction into ‘reverse engineering’ and ‘concept-based interpretability’ (Section 2).
>   * **Associated requested change**:
>     > Expand the methodological summary beyond the reverse engineering vs. concept-based distinction.
>
>   * **Response**:
>
>
>     We agree that the summary would benefit from further explication of the methodological categorisation use in the paper. We now have expanded the summary in the introduction of Section 2 “Open problems in mechanistic interpretability methods and foundations”, in order to give a more detailed account of the categorization.

---

> ### Author Response · Authors · 2025-05-21
> **Changes made in response to reviewer uHkd feedback (cont.)(2/3)**
>
> * **Weakness**:
>   > The focus on the human perspective for ‘concept-based interpretability’ is not clear. Should an interpretability approach not focus on providing any concept, even if not aligned with human perspectives?
>   * **Associated requested change**:
>     > Clarify the role of human vs. non-human concepts in concept-based interpretability.
>
>   * **Response**:
>
>
>     We agree that the ‘human-ness’ of the concepts can be emphasized. We have made the following changes:
>     * In the introduction of section 2, we have added ‘human-derived’ to the following: “Conversely, the second approach, sometimes referred to as `concept-based interpretability', proposes a set of (human-derived) concepts that might be used by the network and then…”
>     * In the section on Concept-based probes, we have added the latter sentence in the following: “A concept-based probe is a classifier trained to predict a concept from the hidden representation of another model \citep{hupkes2018visualisationdiagnosticclassifiersreveal}. These concepts are typically human concepts of special interest.”
>     * In section ‘Probes need carefully chosen data for well-defined concepts’, we have included the following language: “Concept-based probing requires a labeling function that assigns labels for the (human) concept of interest to input data.”
>
> * **Weakness**:
>   > The survey lacks a dedicated discussion and connection to frameworks developed for evaluating explanations in the context of attributions, e.g. how do commonly used evaluation metrics like “faithfulness” and “plausibility” transfer to mech interp? Section 2.3 touches upon these points (see “5. Final Validation”), but only refers to the limited recent works, see, e.g [Hed23, Vil21] for broader reviews.
>   * **Associated requested change**:
>     > Include a more extensive discussion on evaluation frameworks and metrics like faithfulness and plausibility, especially in reference to more than a decade of research in developing faithful attribution approaches.
>
>   * **Response**:
>
>
>     Although our review covers attribution methods, their use in MI, and their weaknesses for the purposes of MI in subsection “Attribution methods are necessary for causal explanations but are often difficult to interpret”, we agree with the reviewer that we did not dedicate discussion to frameworks developed for evaluating explanations. This is in large part because these frameworks for assessing attribution methods are typically adapted to ‘non-MI’ use cases. For instance, the Quantus benchmark quantifies metrics to evaluate explanations of models’ predictions, which are explanations of a models full input-output behavior. These benchmarks are thus adapted to attributions that span from output to input space, or hidden to input space. Attributions in MI are rarely used to attribute output predictions to the input space (though sometimes will attribute hidden activations to the input space); but the primary use is to attribute hidden components with respect to other hidden components. Metrics to quantify explanations such as ‘faithfulness’ and ‘plausibility’ will often therefore be subtly incompatible across the use cases typical in MI. In light of this, we have added Hedstrom et al. (2023) to the section on ‘Validating interpretability methods using benchmarks’ with the following text: “Other benchmarks offer tests for multiple facets of the interpretability pipeline in order to test explanation faithfulness, but may require further adaptation to the context of mechanistic explanations \cite{Hedstrom2023quantus}”

---

> ### Author Response · Authors · 2025-05-21
> **Changes made in response to reviewer uHkd feedback (cont.)(3/3)**
>
> * **Weakness**:
>   > Figure 6 does not convey any relevant visual information as it summarizes the headings of Section 3.
>   * **Associated requested change**:
>     > Remove/revise Figure 6 to convey more meaningful visual information.
>
>
>   * **Response**:
>
>
>     That’s fair - we have removed figure 6.
>
>
>
> * **Weakness**:
>   > What’s the use of discussing and dissecting a fairly recent direction of “sparse dictionary learning” and SAEs that has been advocated for by the mech interpretability community from 2021 onwards, to realize that it comes with many flaws that remain difficult to resolve in the context of deep learning models (see Figure 4). One conclusion from this survey may be that this research direction has not been able to fulfil the promises, what are alternatives that may be able to better reconcile these methodological issues. Sparse coding and compressed sensing have a long and theoretically well-grounded history which should be discussed in this context as many of the encountered issues have been identified previously in other contexts, e.g. source separation. 3.2.1
>
>   * **Associated requested change**:
>     > Discuss alternative methods to sparse dictionary learning and relate to prior work in sparse coding and source separation, should this direction be pursued?
>
>   * **Response**:
>
>
>     We agree with the reviewer that the research direction has so far not been able to fulfil the promises and that better alternatives are needed. This is why our review follows up the account of SDL’s failings with a call for better theoretical foundations and suggest other potential directions that researchers might consider as alternatives to SDL.  SDL is probably just solving the _wrong problem_, whereas adjacent methodologies, such as sparse coding/compressed sensing, are likely to provide better ways of solving the _same problem_. Nonetheless, we agree with the reviewer that more could be done to point toward potential alternatives. We therefore added a sentence "The field should explore very different approaches that ... ".
>
>
>
> * **Weakness**:
>   > Short-sighted survey with respect to undesired black-box behaviors, see e.g. [Lap19, Gei20]
>
>   * **Associated requested change**:
>     > Broaden the discussion on undesired black-box behavior.
>
>   * **Response**:
>
>
>     We agree that undesired black box behaviors were under-represented. We have added some content to address this to the section “Predicting behavior in novel situations“: “Mechanistic interpretability might also be able to identify when networks have learned `shortcuts' \cite{Geirhos_2020} that might lead to generalization failures under distributional shifts, which may result in unexpected and undesirable behavior. Identifying these `shortcuts' in advance may help predict these failures in advance  \cite{Lapuschkin2019}.”
>
>
> * **Weakness**:
>   > The paper lacks a clear prioritization, in my view the main priority should be to ground mech interp in better methodology, yet on this point the paper remains vague and focuses on a summary of all things that have not worked.
>
>   * **Associated requested change**:
>     > Prioritize methodological grounding as a central challenge for the field.
>
>   * **Response**:
>
>
>     Our paper’s main aim is to outline the current frontier of open problems in mech interp, aiming above all for comprehensiveness. We hope this review can serve as an impartial guide that represents as broad a view on the field (and what its priorities should be) as possible. That said, we do not shy away from pointing toward priorities where our numerous authors agree; we do offer some priorities for the field throughout the review, which are best summarized in the conclusion.
>
>
> * **Weakness** (mentioned in related work section):
>   > Section 2.2.3. The notion of “spurious correlations” seems relevant here.
>
>   * **Response**:
>
>
>     We agree that this may be an additional issue. We have added the following text to section 2.2.3: “This approach may also create spurious correlations, an issue shared with concept-based probing (\Cref{subsubsec:concept-based-probes}).”
>
>
> ## Missed related works:
>
> We thank the reviewer for flagging these works. We have included them all in the appropriate sections with comments (see revised paper).

---

### Review · Reviewer_eDJL · 2025-04-18

**Summary Of Contributions:**

The submission is a forward-looking assessment of mechanistic interpretability (MI) research.  At a high level, the first half discusses open problems in MI methodology (Sec 2) while the second half focuses on future applications and societal implications (Secs 3,4).

**Audience:**

Yes

**Broader Impact Concerns:**

No concerns.

**Claims And Evidence:**

Yes

**Requested Changes:**

The weakness above should be address or rebutted, as well as the following more minor points:
- Please explain the term “streetlight interpretability”, which is likely not standard terminology to a broader audience.
- The review defines/frames MI in terms of generalization.  Page 2: “In this review, we use the term ‘mechanistic interpretability’ in a technical sense, referring specifically to work that investigates the mechanisms underlying neural network generalization”, and then multiple times in Sec 2 generalization is positioned as central to MI.  While MI can certainly inform our understanding of generalization, it is a broader pursuit to understand the internal mechanisms of networks.  Please correct me if I’m misreading.
- Nit: page 26, “Current approaches for knowledge discovery from data involve statistical or causal analysis, dimensionality reduction, or using machine learning models that are inherently interpretable. These techniques can be valuable, but are influenced by human priors, typically assume linear relationships between variables, and cannot handle massive multimodal data. **On the other hand, neural networks can do these things.**”  The bolded sentence is awkward if you mean it as neural networks ameliorate all three issues, as “do these things” does not fit with being “influenced by human priors” or “assume linear relationships”.

**Strengths And Weaknesses:**

## Strengths
While there have been many recent reviews/surveys of MI, this one offers a sharp assessment of the current issues and shortcomings on the methodological front.  I found Sec 2, and particularly the frank discussion of issues with SDL, to be insightful and a pleasure to read.

## Weaknesses
In contrast to the sharp and targeted assessment of Sec 2, Secs 3 and 4 were often about implications of understanding models generally instead of MI specifically.  I would recommend zeroing back in on the specific role of MI in these applications and societal implications.

---

> ### Author Response · Authors · 2025-05-21
> **Changes made in response to reviewer feedback**
>
> We thank reviewer eDJL for their assessment and are glad that they found it insightful and a pleasure to read.
>
> ## Responding to weaknesses:
> > In contrast to the sharp and targeted assessment of Sec 2, Secs 3 and 4 were often about implications of understanding models generally instead of MI specifically. I would recommend zeroing back in on the specific role of MI in these applications and societal implications.
>
> We agree with this comment, though disagree that it indicates the suggested action. The extent to which we actually understand models is the extent to which our understanding maps onto their actual internal mechanisms. This map may be fine-grained or course. Anything you can do with a coarse-grained understanding, you can do with a fine-grained understanding and more. We think it is reasonable to characterize MI as aiming for the deepest, most comprehensive understanding of a neural network out of any field that aims to understand neural networks. Therefore, regarding section 3, which outlines various problem areas for the application of mechanistic interpretability, we think that MI does indeed apply evenly to all areas mentioned. Course-grained approaches to understanding networks may play more specific roles in each problem area, but MI should not. We detail throughout section 3 the areas where understanding the mechanisms of neural networks would be helpful in particular applications, and are not sure maximally finegrained, maximally ambitious aims of MI can be zeroed back in the text. An analogous argument applies for section 4 (omitted due to character constraints).
>
> Overall, our argument here is that the consequences of successful MI would in fact touch the very broad range of topics covered in sections 3 and 4, making it difficult to zero back on MI’s specific role. It is other fields that are less ambitious in their aims that would warrant zeroing back, due to their more specific consequences for applications, policy levers, and social consequences. We hope this has addressed the reviewer’s concern. If not, we would be glad to continue the discussion, and think that examples may be helpful for our understanding of the reviewer’s concern.
>
> ## Responding to requested changes:
> > Please explain the term “streetlight interpretability”, which is likely not standard terminology to a broader audience.
>
> We agree this deserves fuller explication. We have now defined the term in the text.
>
>
> > The review defines/frames MI in terms of generalization. ... While MI can certainly inform our understanding of generalization, it is a broader pursuit to understand the internal mechanisms of networks. Please correct me if I’m misreading
>
> Prior to submission, several of the authors discussed this frame in detail before deciding to adopt it.
>
> We decided to adopt it because the idea of a ‘mechanism’ that does not aid with generalization is somewhat paradoxical. We think that understanding the 'internal mechanisms' of networks is therefore equivalent to understanding the mechanisms of their generalization. We provide a few intuition pumps to help convey the intuitions behind this frame:
>  - We can’t really do MI on a randomly initialized network. This would simply involve tracing each datapoint through the network, and any structure that emerges would be random projections of structure in the dataset (which isn’t the main object of study of mech interp).
> - A concern might be “What about mechanisms that serve only memorization?”. Memorization is arguably just a special case of generalization: Memorization is just good performance on a very specific subset of the data distribution.
> - If there is structure in a neural network beyond its random initialization, it is there because optimization in pursuit of the training objective put it there. Any non-random structure in a neural network is there in service of performance. Mechanistic interpretability is really the pursuit of understanding of modularity of this structure -- structure that are ultimately either random or in pursuit of generalization.
> - We will know if MI succeeds when it gives us insight (or, concretely, successful predictions) about ways in which networks generalize.
> - Suppose a paper published (to pick a random topic) a complicated analysis of the patterns of prime numbers in the weight matrices of a neural network. This analysis would indeed be an analysis of the structure of the internals of the network. One might therefore argue that it therefore concerns the network’s ‘mechanisms’; but this definition feels satisfying because it is not a study of the network’s structure with reference to how the network actually performs its function, where its function is ‘to do well on the training/test set’.
>
>
> > Nit: page 26, “Current approaches ... or “assume linear relationships”.
>
> Good point. It is now amended.

---

### Review · Reviewer_ZeP2 · 2025-05-08

**Summary Of Contributions:**

This piece is a thorough review of the current state and potential future of mechanistic interpretability, including its implications across other disciplines. It cites a wealth of relevant literature.

**Audience:**

Yes

**Broader Impact Concerns:**

Implications are appropriately addressed, especially when warning against over stating what mech-interp is capable of.

**Claims And Evidence:**

Yes

**Requested Changes:**

I think it is crucial that the authors give some space to the possibility that some of the goals of mech-interp may simply be impossible to achieve. Especially when describing future potential uses of mech-interp, we need to accept that current challenges might actual reflect inherent difficulties in understanding a large distributed system that will not be resolved by better methodology. Perhaps put another way, even if we can "carve the model at its joints" it may still not be in a very usable shape for us. More on this can be found here: https://arxiv.org/abs/1907.06374

I think the authors are also missing an additional interdisciplinary benefit of advances in mech-interp which is that it can help neuroscientists decide how to analyze real neural networks. Methods won't be able to be directly ported, but certainly concepts and approaches can be shared. More on this here:  https://www.sciencedirect.com/science/article/pii/S1389041723000906 I think this would be a strong addition to the piece.

The authors may want to add this recent work to their references. I don't think it changes any of the core claims, but does represent a lot of them: https://transformer-circuits.pub/2025/attribution-graphs/methods.html

Maybe explain or give more examples of attribution methods after this: "Attribution methods are necessary for causal explanations but are often difficult to interpret"

word missing: "networks is that the latents identified by depend on the data set used to train them."

**Strengths And Weaknesses:**

Strengths:

Writing is clear

Coverage is thorough

Several important though usually under-discussed ideas are highlighted including: the need for validation, the use of 'model organisms', and the need for more diverse model types

Weaknesses:

It is long. Though on the whole it is not terribly repetitive or long-winded, so this may not be a weakness.

---

> ### Author Response · Authors · 2025-05-21
> **Changes made in response to reviewer feedback**
>
> We thank reviewer ZeP2 for their engagement with our work and are grateful that they found its coverage thorough and clear.
>
> We are also glad to hear that they appreciated our discussion of several under-discussed ideas in mech interp.
>
> Regarding each of their requested changes. We have addressed each in the text. We discuss each change in light of each request below:
>
> > I think it is crucial that the authors give some space to the possibility that some of the goals of mech-interp may simply be impossible to achieve. Especially when describing future potential uses of mech-interp, we need to accept that current challenges might actual reflect inherent difficulties in understanding a large distributed system that will not be resolved by better methodology. Perhaps put another way, even if we can "carve the model at its joints" it may still not be in a very usable shape for us. More on this can be found here: https://arxiv.org/abs/1907.06374
>
> We agree with the reviewer that this is a major concern in the field.
>
> We have added a paragraph highlighting this concern at the end of section 2.1 “While progress toward the goals of reverse engineering continues, it must be recognized that it is an ambitious goal, especially for very large neural networks. It may be the case that the field fails ultimately to make progress toward complete reverse engineering because of the fundamental difficulty of understanding such large, complex systems. One manifestation of this might be that, even though we might make progress toward `carving networks at their joints', the components that result are either too complex or numerous to make ready use of them. Similar concerns have motivated some researchers in computational neuroscience to aim to study the learning process of a neural system, rather than the mechanisms that result \cite{lillicrap2019doesmeanunderstandneural}.”
>
> We have also added an addendum to the first sentence of the conclusion, which now reads “While mechanistic interpretability has made meaningful progress in both methods and applications, significant challenges remain before we can achieve many of the field’s ambitious goals, **if they can be achieved**.”
>
> > I think the authors are also missing an additional interdisciplinary benefit of advances in mech-interp which is that it can help neuroscientists decide how to analyze real neural networks. Methods won't be able to be directly ported, but certainly concepts and approaches can be shared. More on this here: https://www.sciencedirect.com/science/article/pii/S1389041723000906 I think this would be a strong addition to the piece.
>
> We agree with the reviewer that this is an exciting area of interdisciplinary interaction, and indeed several of the authors share this motivation for entering the field! There is an unusual amount of potential for cross-talk between the two fields. We have therefore added a sentence in the introduction:
>
> “What new laws of nature governing the mechanisms of minds might we discover from studying the internal workings of AI systems? _What new methods for analyzing biological neural systems might neuroscientists be able to glean from studying artificial ones?_ ”
>
> It is perhaps worth justifying why we have not added a full section on this topic further to this addition. The review is not aiming to catalogue the benefits of mech interp for other fields. It aims to identify open problems in mech interp. The reviewer’s suggestion is less of an ‘open problem in mech interp’ and more of an ‘open problem in computational neuroscience’, even though we very much agree with the importance of interdisciplinary interaction between these two disciplines.
>
> > The authors may want to add this recent work to their references. I don't think it changes any of the core claims, but does represent a lot of them: https://transformer-circuits.pub/2025/attribution-graphs/methods.html
>
> We have now added this recent reference to the paper in multiple places and noted the progress that it represents.
>
>
> > Maybe explain or give more examples of attribution methods after this: "Attribution methods are necessary for causal explanations but are often difficult to interpret"
>
> We have now added specific mention of several attribution methods.
>
> “Specific examples of attribution methods include Grad-CAM \citep{Selvaraju_2019_GradCAM} and Integrated Gradients \citep{Sundarajan_2017_IntegratedGradients}, as well as perturbation or model-agnostic approaches such as LIME \citep{Ribiero_2016_LIME} and SHAP \citep{Lundberg_2017_SHAP}.”
>
>
> > word missing: "networks is that the latents identified by depend on the data set used to train them."
>
> Thank you - we have amended the sentence now to read “A complicating factor in using SDL to identify the learned mechanisms of neural networks is that the latents identified by SDL depend heavily on the data set used to train them.”

---

> > ### Comment · Reviewer_ZeP2 · 2025-06-03
> >
> > Sorry, I should've clarified that I meant the neuroscience connection may be relevant in a similar way as what is discussed in "Using mechanistic interpretability for ‘microscope AI’", not as any core part of the review or any new section. But it is of course a decision for the authors.
> >
> > I am happy with the work and think it is a very nice contribution.

---

### Review · Reviewer_1PFg · 2025-05-15

**Summary Of Contributions:**

The paper is an extensive review of mechanistic interpretability. It covers a large range of publication and it contributes by creating a solid foundation of literature that describes the approaches, limitations and existing challenges in the field. -

**Audience:**

Yes

**Broader Impact Concerns:**

No main concern

**Claims And Evidence:**

Yes

**Requested Changes:**

Following the weaknesses, I suggest:
- revising the paper structure, enhancing the level of abstraction and finding connections between the different lines of works, or putting them in comparison side by side with the aid of tables and other visualizations
- improving all the figures by clarifying the legend, ensuring that they are visible and detailed enough to provide high quality information to the reader
- Enhance the discussion on the need of interpretability or the possibility that we will have to live with an uninterpretable boundary

The authors could consider extending the references to the discovery of concepts, e.g.:
- how it ties to superposition and polysemanticity:
Graziani, M., O'Mahony, L., Nguyen, A. P., Müller, H., & Andrearczyk, V. Uncovering Unique Concept Vectors through Latent Space Decomposition. Transactions on Machine Learning Research.
- how it can be used for model editing:
Oldfield, J., Tzelepis, C., Panagakis, Y., Nicolaou, M., & Patras, I. PandA: Unsupervised Learning of Parts and Appearances in the Feature Maps of GANs. In The Eleventh International Conference on Learning Representations.
- how it can be obtained and iteratively refined with NMF:
Fel, T., Picard, A., Bethune, L., Boissin, T., Vigouroux, D., Colin, J., ... & Serre, T. (2023). Craft: Concept recursive activation factorization for explainability. In Proceedings of the IEEE/CVF Conference on Computer Vision and Pattern Recognition (pp. 2711-2721).

Additionally, it would also be worth mentioning the work on meta-evaluations:
- Hedström, Anna, et al. "Quantus: An explainable ai toolkit for responsible evaluation of neural network explanations and beyond." Journal of Machine Learning Research 24.34 (2023): 1-11.

**Strengths And Weaknesses:**

Strengths:
- The formalisation of the topics in interpretability research is a relevant need in the community, hence the paper fills an important gap
- The literature review covers a broad range of existing work
- The formalisation of sparse dictionary learning is interesting and relevant, and so are the concepts of "imperfect interpretability" and "model organisms".

Weaknesses:
- The structure of the paper can be vastly improved as right now it reminds of a collection of summaries. As of now, the authors mention the main publications and research directions with endless lists of bullet points and mini paragraphs. While I see the struggle of handling such a large amount of literature, they could have done a better job at summarising, grouping and collecting works in collective views, literature maps and summary tables. For example, point 1 on page 18 actually generalises to many of the other methods that have been described in the previous pages, but here it is merely presented as a component of only one of them. What is the point of compiling a literature review if no such level of abstraction is introduced to simplify the understanding of the readers on a given topic?
- The figures are very poor quality. Figure 2 is just white. Figure 6 is broken. Apart from these two, I find the other ones hard to interpret as there is no legend of the symbolism being used, and a lot is left to intuition. This can be improved.
-  The discussion lacks depth if there is no argument against the need for interpretability. There may be a part of the model that will always remain impossible to interpret, as there is a misalignment between how humans and machine works, and there will only be. I fear this paper is missing the opportunity to shift the discussion on whether there is the need for "perfect interpretability", which may as much be unachievable. As in most engineering systems, one mainly cares about the safe boundary or margin, which ensure smooth and safe functioning of the system, rather than a perfect functioning. A similar concept could have been expanded in the context of "imperfect interpretability", argumenting on what and how such safe margin could be defined and tested by future research. The authors also touch on it in figure 5, highlighting "irrelevant components", but fail to describe this concept further and how does it tie to assessing their relevance in the model (e.g. to pruning) and on whether they could, at times (if pruning is followed by a degradation in  performance) they may represent what is left "uninterpretable" in the model.

---

> ### Author Response · Authors · 2025-05-21
> **Changes made in response to reviewer feedback**
>
> We thank the reviewer for their engagement with our paper and are glad to hear that our work fills an important gap.
>
> We respond to the weaknesses and suggested changes below.
>
> ## Regarding paper structure:
> > The structure of the paper can be vastly improved as right now ... [cont.] ... literature maps and summary tables
>
> One of the unfortunate things about the TMLR default style is that it precludes tables of content (we hope a later version may be permitted to include this), which would make the rationale behind the current structure more apparent. In lieu of a table of contents, we have introduced a summary figure (new Fig 1).
>
> In addition, when rereading our review with this feedback in mind, we saw a further opportunity to introduce helpful structure to the manuscript. We have added a summary table of the methods in section 2, providing an at-a-glance summarisation of the methods used in the field and their open problems.
>
> ## Figures
>
> > The figures are very poor quality. Figure 2 is just white. Figure 6 is broken. Apart from these two, ... [cont.] ...  a lot is left to intuition.
>
> We are sorry to hear this about the figures. We suspect that at least for figures 2 and 6, this may be due to technical issues on the reviewer’s side. For the authors, figure 2 is not white, and figure 6 is not broken. We’d ask the reviewer to confirm if the problem persists when viewing the paper pdf in another browser. In the meantime, we’ll investigate the potential cause of this suspected formatting issue in case it is a wider problem.
>
> To address the reviewer’s other concerns about the figures, we have added further labels for symbols in the updated versions of the figures.
>
>
> ## Other comments
>
> > The discussion lacks depth if there is no argument against the need for interpretability. There may be a part of the model that will always remain impossible to interpret ...
>
> We agree with the reviewer that this is a major concern in the field. We have added a paragraph highlighting this concern at the end of section 2.1. See paragraph starting and ending with: “While progress toward the goals of reverse engineering continues, it must be recognized that it is an ambitious goal, ... the mechanisms that result \cite{lillicrap2019doesmeanunderstandneural}.”
>
> We have also added an addendum to the first sentence of the conclusion, which now reads “... significant challenges remain before we can achieve many of the field’s ambitious goals, **if they can be achieved**.”
>
>
> > As in most engineering systems, one mainly cares about the safe boundary or margin, which ensure smooth and safe functioning of the system, rather than a perfect functioning.  ... [cont.] ...  what is left "uninterpretable" in the model.
>
> We are not confident we understand the reviewer’s argument, so we will do our best to recapitulate the version of it that we understood so that the reviewer can correct us if we have misunderstood them:
>
> Our version of what the reviewer is arguing: MI may not need to be perfect in order to be useful. Some parts of the model might not be useful for computation and hence can be pruned. But these parts may represent what is ‘uninterpretable’ in the model.
>
> Our response to that version of what the reviewer is arguing: In section 2.1.3 (‘causal interventions’) we cite a number of articles related to pruning, “Other work learns masks over network components to remove irrelevant components (De Cao et al., 2020; Csordás et al., 2021; Davies et al., 2023)” and therefore agree with the reviewer that it is likely that some parts of the model can likely be pruned. We also think it is likely that those parts are the most likely to be ‘uninterpretable’, precisely because they appear to do no useful computation. However, we don’t understand the connection the reviewer seems to be making between this line of work and the idea that MI may not need to be perfect in order to be useful (a claim we also agree with, for what it’s worth). We would appreciate if the reviewer could correct our account of their argument and help us understand, especially if the changes we made above regarding the potential impossibility of some of the goals of MI are insufficient.
>
>
>
> ## Paper suggestions:
>
> We thank the author for these references. They are good suggestions. We have included them all:
>
> * Paper: graziani2023uncoveringuniqueconceptvectors
>   * Amendment: In section 2.1.2 we have included a line where we discuss dimensionality reduction methods
> * Paper: oldfield2023pandaunsupervisedlearningparts
>   * Amendment: In section 2.1.2 we have included mention of tensor factorization in the list of methods.
> * Paper: fel2023craftconceptrecursiveactivation
>   * Amendment: In section 2.1.2 we have added Fel et al. (2023) to the list of papers that use NMF for concept discovery/decomposition.
> * Paper: Hedstrom2023quantus
>   * Amendment: A line has been added in the section ‘Validating interpretability methods using benchmarks’.

---

> > ### Comment · Reviewer_1PFg · 2025-06-04
> > **Thanks for the changes**
> >
> > Hi,
> >
> > Thank you for the changes.
> > Indeed, the figure issue was on my end and it was sufficient to download the file with a different browser (weird!).
> >
> > As for the point in my review that was left unclear, I agree with your observations. What I meant referred more to the point of view expressed here: https://arxiv.org/pdf/1610.01256.
> > Quoting the paper:
> >
> > "In mechanical systems, a safety factor is a ratio between the maximal load that does not lead to failure and the load for which the system was designed. Similarly, the safety margin is the difference between the two."
> >
> > And then the authors go forward in explaining how this can be used to constraint the relative risk of harm of a prediction to a maximum value which is still acceptable (safe). I find this concept very interesting from a practical standpoint as it may be a viable way to deal with what may necessarily remain unexplainable in models. This is only a suggestion and the authors do not have necessarily to expand on this aspect, but I thought it would add value to their work.

---

### Decision · Action_Editor_6Fpq · 2025-09-01

**Recommendation:** Accept as is

**Additional Comments:**

This is a timely survey on mechanistic interpretability, highlighting the open problems.

**Audience:**

Yes

**Audience Explanation:**

Mechanistic interpretability (MI) is a growing subfield of research in ML, and I think this article will be useful for everyone working in MI and for researchers who want to enter the field.

**Claims And Evidence:**

Yes

**Claims Explanation:**

This is a review paper on the current state of mechanistic interpretability and open problems that are worth exploring. Three out of four reviewers agree that this is a good review of the field. While the fourth reviewer has concerns about the overfocus on SAEs, I think it is ok given that SAEs are the most popular methods for mechanistic interpretability for the time being.

---

> ### Author Response · Authors · 2025-09-09
> **Thank you to the Reviewers and Action Editor**
>
> Dear Reviewers and Action Editor,
>
> We are delighted that our paper has been accepted for publication at TMLR. We are grateful for your efforts evaluating the paper and highlighting areas for improvement, which has strengthened the paper. We have submitted the final version of the paper.
>
> Thank you again,
> First author, on behalf of co-authors.